# Likelihood over Estimation: Robust Quadratic Discriminant Analysis for Heavy-Tailed Distributions with Theory and Evidence

**Niranjana Ambadi** [1]   **Eugene Pinsky** [2]

## Abstract

Quadratic Discriminant Analysis (QDA) assumes Gaussian class-conditional distributions, causing systematic misclassification when data exhibit heavy tails. We propose Stable-QDA, which replaces the Gaussian likelihood with a symmetric $\alpha$-stable likelihood that decays polynomially rather than exponentially in Mahalanobis distance. Crucially, we find that correcting likelihood misspecification yields larger gains than robustifying parameter estimation: standard estimators (sample mean, Ledoit–Wolf covariance) often outperform robust alternatives when class heteroscedasticity is discriminative. We provide consistency guarantees under infinite-variance regimes, data-driven diagnostics for estimator selection, and demonstrate 15–53% error reduction on real-world heavy-tailed benchmarks.

## 1. Introduction

Quadratic Discriminant Analysis (QDA) is a foundational method in statistical classification, valued for its interpretability, calibrated probabilistic outputs, and computational efficiency. By modeling each class through an explicit generative distribution, QDA yields closed-form training procedures, $\mathcal{O}(Kp^2)$ prediction complexity, and decision boundaries that can be directly inspected and analyzed. These properties make QDA particularly appealing in scientific and high-stakes applications where transparency, uncertainty quantification, and reliability are essential.

Despite these advantages, classical QDA is highly sensitive to violations of its core modeling assumption: Gaussian class-conditional distributions. In many real-world domains, data exhibit pronounced heavy-tailed behavior(Hodgkinson et al., 2025) rather than exponential decay. Network traffic measurements follow power laws due to heterogeneous user behavior and attack dynamics; financial returns and transaction amounts are well modeled by stable or stable-like distributions; and astrophysical signal detection involves distinguishing weak signals from heavy-tailed noise processes. In such settings, observations in the distributional tails are not rare anomalies but often the most informative instances. Gaussian QDA, whose likelihood decays exponentially in the squared Mahalanobis distance, systematically underweights these observations, assigning them vanishing likelihood and inducing distorted decision boundaries that lead to systematic misclassification in the tails.

A common response to this problem is to replace the sample mean and covariance with robust alternatives such as the spatial median, Tyler's M-estimator, or rank-based scatter estimators. While effective under contamination models (Huber & Ronchetti, 2009), these methods leave the Gaussian likelihood unchanged, addressing estimator instability but not likelihood misspecification. We demonstrate that this distinction is crucial: across a wide range of heavy-tailed regimes, correcting the likelihood mismatch yields larger and more consistent improvements in classification performance than robustifying parameter estimation alone.

These observations motivate *Stable-QDA*, a simple and principled extension of classical QDA that replaces the Gaussian likelihood with a symmetric $\alpha$-stable likelihood while retaining well-conditioned estimators for location and dispersion. The $\alpha$-stable family generalizes the Gaussian distribution through a stability index $\alpha \in (0, 2]$, with smaller values corresponding to heavier tails. By the generalized central limit theorem, $\alpha$-stable distributions are the only possible non-degenerate limits of normalized sums of independent random variables, making them a natural and theoretically grounded model for heavy-tailed data. Gaussian QDA is recovered as the special case $\alpha = 2$.

The key insight underlying Stable-QDA is that QDA decision boundaries depend on *likelihood ratios*. Estimation errors that affect all classes similarly tend to cancel in these ratios, whereas likelihood misspecification induces systematic distortions that do not. By replacing exponential tail decay with polynomial decay, the stable likelihood prevents

---

[1]Founder, LawMate Private Limited, North Plains, USA
[2]Associate Professor of the Practice, Department of Computer Science, Boston University Metropolitan College, Boston, USA. Correspondence to: Niranjana Ambadi <chirpybits@gmail.com>.

*Proceedings of the $43^{rd}$ International Conference on Machine Learning*, Seoul, South Korea. PMLR 306, 2026. Copyright 2026 by the author(s).

the over-penalization of extreme but informative observations. Importantly, we show that Stable-QDA is remarkably insensitive to misspecification of the tail index: a fixed choice $\alpha = 1.5$ performs robustly across a wide range of heavy-tailed regimes, eliminating the need for costly tuning.

**Contributions**  Our contributions are: *(i) Stable-QDA:* a novel $\mathcal{O}(Kp^2)$ classifier using $\alpha$-stable likelihoods that remains consistent under infinite variance. *(ii)* We show that correcting likelihood misspecification yields larger gains than robust estimation alone, and characterize when robust estimators help or harm. *(iii)* We propose data-driven diagnostics (along with an open source tool) for estimator selection based on tail heaviness, class heteroscedasticity. *(iv)* We demonstrate how Stable-QDA improves error rates and tail-conditional recall on heavy-tailed benchmarks while matching or exceeding Gaussian QDA on accuracy and PR-AUC.

Together, these results suggest a shift in emphasis for heavy-tailed classification: correctly specifying the likelihood is often more important than aggressively robustifying parameter estimation. While discriminative classifiers (neural networks, random forests, XGBoost, kNN, etc.) are often preferred for their predictive power, recent work revisiting the generative-discriminative trade-off (Zheng et al., 2023) suggests that generative models remain superior in low-sample regimes and under specific structural constraints, provided the class-conditional distributions are well-specified. Our work addresses a major gap in this specification for heavy-tailed data. Stable-QDA preserves the interpretability and efficiency of classical QDA while correcting its most fundamental failure mode under heavy-tailed data.

## 2. Related Work

**Robust discriminant analysis.** Classical discriminant analysis is sensitive to outliers, motivating robust alternatives using M-estimators for location (Huber, 1964) and high-breakdown covariance estimators such as MCD (Hubert & Van Driessen, 2004). Recent work incorporates robust estimators into generalized QDA (Ghosh et al., 2021) or flexible discriminant frameworks (Houdouin et al., 2022). However, these methods retain Gaussian-based decision rules, limiting effectiveness for genuinely heavy-tailed data.

**Elliptical models for classification.** Several works consider discriminant rules based on elliptically symmetric distributions. Bose et al. (2002) proposed generalized QDA (GQDA), adapting the Mahalanobis distance via a tail-dependent exponent but without an explicit generative likelihood. Shen & Feng (2025) introduced spatial-sign QDA (SSQDA), achieving optimal convergence rates under elliptical models but without specifying class-conditional densities. In contrast, we model each class with symmet-

ric $\alpha$-stable distributions, yielding a likelihood-based discriminant that remains well-defined under infinite variance ($\alpha < 2$) with Bayes consistency guarantees.

**$\alpha$-stable distributions.** The theory of stable distributions is well established (Samorodnitsky & Taqqu, 1994; Nolan, 2020), with applications in finance (Rachev & Mittnik, 2000), signal processing (Nikias & Shao, 1995), and heavy-tailed regression (Lugosi & Mendelson, 2023). General treatments of multivariate stable distributions appear in Fang et al. (2018) and Gupta & Varga (1993). We leverage this theory for discriminant analysis, connecting stable likelihoods with quadratic decision rules.

**Heavy-tailed and imbalanced classification.** Methods such as Isolation Forest (Liu et al., 2008) and deep one-class models (Ruff et al., 2018) detect deviations from nominal data but lack calibrated probabilities or interpretable boundaries. Stable-QDA provides a complementary generative approach preserving probabilistic interpretability.

**Dispersion estimation under heavy tails.** Tyler's M-estimator (Tyler, 1987) provides distribution-free consistent shape estimation, extended to regularized settings (Pascal et al., 2008). Rank-based alternatives combine robust scales with correlation estimation (Boudt et al., 2012; Maronna et al., 2019). Despite strong robustness guarantees, we find these estimators often underperform shrinkage-based estimators like Ledoit–Wolf (Ledoit & Wolf, 2004) in classification when class scales differ. While recent work at ICML continues to refine robust estimators for heavy-tailed mean estimation and bandits, we argue that for classification, the primary bottleneck is likelihood misspecification (Høgsgaard & Paudice, 2025; Lee et al., 2020; Huang et al., 2022).

## 3. Background

### Quadratic Discriminant Analysis

Quadratic Discriminant Analysis (QDA) is a generative classifier that models each class $k \in \{1, \ldots, K\}$ with a multivariate Gaussian distribution. Given training data $\{(\mathbf{x}_i, y_i)\}_{i=1}^n$ where $\mathbf{x}_i \in \mathbb{R}^p$ and $y_i \in \{1, \ldots, K\}$, QDA estimates class-specific means $\hat{\boldsymbol{\mu}}_k$ and covariances $\hat{\boldsymbol{\Sigma}}_k$ via maximum likelihood. By Bayes rule: $\hat{y}(\mathbf{x}) = \arg\max_k \left\{ \log \pi_k - \frac{1}{2} \log |\hat{\boldsymbol{\Sigma}}_k| - \frac{1}{2} D_k(\mathbf{x}) \right\}$ where $\pi_k = P(y = k)$ is the class prior and $D_k(\mathbf{x}) = (\mathbf{x} - \hat{\boldsymbol{\mu}}_k)^\top \hat{\boldsymbol{\Sigma}}_k^{-1} (\mathbf{x} - \hat{\boldsymbol{\mu}}_k)$ is the squared Mahalanobis distance. The resulting decision boundaries are quadratic surfaces when class covariances differ.

QDA offers closed-form parameter estimation, $O(Kp^2)$ prediction complexity, and calibrated probabilistic outputs. However, its Gaussian likelihood decays exponentially in $D_k(\mathbf{x})$, systematically underweighting observations in the

distributional tails. When data are heavy-tailed, this mismatch causes extreme-but often informative-observations to receive vanishing likelihood, distorting decision boundaries and inducing systematic misclassification precisely where correct classification matters most.

**Heavy Tails in Classification.** Heavy-tailed distributions arise naturally in many classification domains. Network traffic exhibits power-law behavior due to the diversity of protocols, user behaviors, and attack patterns. Financial data-returns, transaction amounts, interarrival times-follow stable or stable-like distributions with $\alpha$ typically between 1.4 and 1.9 (Rachev & Mittnik, 2000). Scientific measurements often contain heavy-tailed noise from instrumental effects or environmental interference. In these domains, observations in the distributional tails are often the most consequential: anomalous network packets may indicate intrusions, extreme transactions may signal fraud, and faint signals may represent genuine discoveries.

The standard response to heavy-tailed data is to robustify parameter estimation using methods such as the spatial median for location and Tyler's M-estimator (Tyler, 1987) for scatter. While these estimators provide strong guarantees under contamination models, they leave the Gaussian likelihood unchanged and thus address estimator instability but not likelihood misspecification. Our empirical results (Section 7) reveal that this distinction is crucial: correctly specifying the likelihood yields larger and more consistent classification improvements than robust estimation alone.

**$\alpha$-Stable Distributions.** The $\alpha$-stable family generalizes the Gaussian distribution to accommodate heavy tails. A random variable $X$ follows a symmetric $\alpha$-stable distribution, denoted $X \sim S_\alpha(\gamma, \delta)$, if its characteristic function is:

$$\phi_X(t) = \exp\left(i\delta t - \gamma^\alpha |t|^\alpha\right), \tag{1}$$

where $\alpha \in (0, 2]$ is the *stability index* (tail parameter), $\gamma > 0$ is the *scale parameter*, and $\delta \in \mathbb{R}$ is the *location parameter*. The stability index $\alpha$ controls tail heaviness: smaller $\alpha$ implies heavier tails. When $\alpha = 2$, we recover the Gaussian distribution; when $\alpha = 1$, we obtain the Cauchy distribution.

A key property of $\alpha$-stable distributions with $\alpha < 2$ is that they have *infinite variance*. While the sample covariance is not a consistent estimator of the dispersion matrix under $\alpha < 2$, classification depends only on relative Mahalanobis distances across classes. Shrinkage regularization stabilizes these distances sufficiently for accurate discrimination, even when absolute parameter consistency fails.

**Multivariate extension.** For multivariate classification, we consider elliptically contoured $\alpha$-stable distributions. A random vector $\mathbf{X} \in \mathbb{R}^p$ follows an elliptical $\alpha$-stable distribution with location $\boldsymbol{\mu}$ and dispersion $\boldsymbol{\Sigma}$ if:

$$\mathbf{X} \overset{d}{=} \boldsymbol{\mu} + A^{1/2}\boldsymbol{\Sigma}^{1/2}\mathbf{U}, \tag{2}$$

where $A$ is a positive $\alpha/2$-stable random variable and $\mathbf{U}$ is uniformly distributed on the unit sphere in $\mathbb{R}^p$. The dispersion matrix $\boldsymbol{\Sigma}$ determines the shape of equidensity contours, analogous to the covariance matrix for Gaussians.

**Why Gaussian QDA fails.** For symmetric $\alpha$-stable distributions with $\alpha \in (1, 2)$, the mean exists but the second moment is infinite. The sample covariance diverges, destabilizing Mahalanobis distances, while the sample mean converges at the slower rate $n^{1/\alpha - 1}$ rather than $n^{-1/2}$. More fundamentally, the Gaussian likelihood decays exponentially in squared Mahalanobis distance, systematically underweighting tail observations.

## 4. Method

We present Stable-QDA, a quadratic discriminant classifier designed for heavy-tailed data. Our approach retains the computational simplicity and interpretability of classical QDA while replacing the Gaussian likelihood with an $\alpha$-stable likelihood that accommodates heavy tails.

### 4.1. Problem Setup

Consider a $K$-class classification problem with training data $\{(\mathbf{x}_i, y_i)\}_{i=1}^n$, where $\mathbf{x}_i \in \mathbb{R}^p$ and $y_i \in \{1, \ldots, K\}$. Classical QDA assumes class-conditional Gaussian distributions:

$$\mathbf{x} \mid y = k \sim \mathcal{N}(\boldsymbol{\mu}_k, \boldsymbol{\Sigma}_k), \tag{3}$$

and classifies via the Bayes rule:

$$\hat{y}(\mathbf{x}) = \arg\max_k \left\{\log \pi_k + \log f_k(\mathbf{x})\right\}, \tag{4}$$

where $\pi_k = P(y = k)$ is the class prior and $f_k$ is the class-conditional density. When data exhibit heavy tails–the Gaussian assumption is violated, leading to degraded classification performance, particularly for observations in the distributional tails.

### 4.2. Stable-QDA: Model and Likelihood

**Likelihood ordering under elliptical $\alpha$-stable models.** For an elliptically contoured distribution with location $\mu$ and dispersion $\Sigma$, the class-conditional density admits the generic form

$$f(x) = |\Sigma|^{-1/2} g_\alpha(D(x)), \tag{5}$$

where $D(x) = (x - \mu)^\top \Sigma^{-1}(x - \mu)$ is the squared Mahalanobis distance and $g_\alpha : \mathbb{R}_+ \to \mathbb{R}_+$ is the radial density determined by the stability index $\alpha$ (Samorodnitsky & Taqqu,

1994; Nolan, 2020). For symmetric $\alpha$-stable distributions, $g_\alpha$ is strictly decreasing in $D$.

Crucially, Bayes-optimal classification depends only on the *ordering* of class-conditional likelihoods across classes. Therefore, any monotone transformation of $g_\alpha(D)$ yields an equivalent discriminant rule with the same decision boundary. This observation allows us to replace the intractable stable density with a tractable, order-equivalent surrogate that preserves the polynomial tail decay characteristic of $\alpha$-stable laws.

### 4.2.1. DERIVATION OF THE SURROGATE LIKELIHOOD

For elliptically contoured $\alpha$-stable distributions, the density admits the stochastic representation (Samorodnitsky & Taqqu, 1994):

$$X \mid Y = k \stackrel{d}{=} \mu_k + A_k^{1/2}\Sigma_k^{1/2}U \qquad (6)$$

where $U$ is uniformly distributed on the unit sphere in $\mathbb{R}^p$ and $A_k \sim S_{\alpha/2}(1,1,0)$ is a positive $\alpha/2$-stable random variable independent of $U$.

The density can be written as:

$$f_k(x) = |\Sigma_k|^{-1/2}\int_0^\infty r^{-p}h_{\alpha/2}(r)\exp\left(-\frac{D_k(x)}{2r}\right)dr \qquad (7)$$

where $h_{\alpha/2}$ is the density of the $\alpha/2$-stable subordinator and $D_k(x) = (x - \mu_k)^\top\Sigma_k^{-1}(x - \mu_k)$.

For large $D_k$, the dominant contribution comes from large $r$, where $h_{\alpha/2}(r) \sim r^{-\alpha/2-1}$ (Zolotarev, 1986). This yields the tail approximation:

$$f_k(x) \sim |\Sigma_k|^{-1/2}\cdot D_k(x)^{-(\alpha+p)/2} \quad \text{as } D_k(x) \to \infty \qquad (8)$$

Taking logarithms:

$$\log f_k(x) = -\frac{1}{2}\log|\Sigma_k| - \frac{\alpha+p}{2}\log D_k(x) + O(1) \qquad (9)$$

For numerical stability and to handle finite $D_k$, we use the regularized form:

$$\log f_k(x) \approx -\frac{1}{2}\log|\Sigma_k| - \frac{\alpha+p}{2}\log(1+D_k(x)) + C_\alpha \qquad (10)$$

The constant $C_\alpha$ cancels in likelihood ratios, so we omit it. The key property is that $\arg\max_k \log f_k(x)$ is preserved by this transformation since $\log(1+D)$ is monotone increasing in $D$.

We model each class with an elliptical $\alpha$-stable distribution:

$$\mathbf{x} \mid y = k \sim S_\alpha(\boldsymbol{\mu}_k, \boldsymbol{\Sigma}_k), \qquad (11)$$

where all classes share a common tail index $\alpha$ but have class-specific location $\boldsymbol{\mu}_k$ and dispersion $\boldsymbol{\Sigma}_k$.

Unlike the Gaussian case, the density of a multivariate $\alpha$-stable distribution lacks a closed form for $\alpha \neq 2$. However, for classification, we require only a function that preserves the *ordering* of likelihoods across classes. We use the following order-equivalent log-likelihood surrogate:

$$\log f_k(\mathbf{x}) \approx -\frac{\alpha+p}{2}\log(1+D_k(\mathbf{x})) - \frac{1}{2}\log|\boldsymbol{\Sigma}_k| + C_\alpha, \qquad (12)$$

where $D_k(\mathbf{x}) = (\mathbf{x} - \boldsymbol{\mu}_k)^\top\boldsymbol{\Sigma}_k^{-1}(\mathbf{x} - \boldsymbol{\mu}_k)$ is the squared Mahalanobis distance to class $k$, and $C_\alpha$ is a normalization constant that cancels in classification.

**Comparison to Gaussian.** The Gaussian log-likelihood is:

$$\log f_k^{\text{Gauss}}(\mathbf{x}) = -\frac{1}{2}D_k(\mathbf{x}) - \frac{1}{2}\log|\boldsymbol{\Sigma}_k| - \frac{p}{2}\log(2\pi). \quad (13)$$

The critical difference is how Mahalanobis distance enters: linearly for Gaussian ($-\frac{1}{2}D_k$) versus logarithmically for stable ($-\frac{\alpha+p}{2}\log(1+D_k)$). For large $D_k$, Gaussian likelihood decays exponentially ($\propto e^{-D_k/2}$) while stable likelihood decays polynomially ($\propto D_k^{-(\alpha+p)/2}$). This prevents over-penalization of tail observations—the fundamental failure mode of Gaussian QDA under heavy tails.

### 4.3. Parameter Estimation

Stable-QDA requires estimating a location vector and a dispersion matrix for each class. For heavy-tailed data, a natural choice is to employ robust estimators, such as the spatial median for location and Tyler's M-estimator for dispersion, which enjoy strong theoretical guarantees including high breakdown points and consistency without finite moments.

However, Tyler's M-estimator is defined only up to scale: it estimates the *shape* of the scatter matrix while normalizing to have fixed trace. In QDA, scale differences between classes are discriminative—the likelihood includes a log-determinant term that penalizes assigning observations to high-variance classes. Consequently, Tyler's trace normalization—while statistically principled—removes scale information that QDA explicitly leverages through its log-determinant term.

**Why not Tyler for shape + robust scale?** A natural alternative is combining Tyler's shape estimate with a robust scale estimator (e.g., MAD). While this hybrid improves over Tyler alone, it still underperforms standard estimators in our classification experiments (Appendix D); the additional estimation step introduces noise without recovering the full benefit of direct scale estimation.

**Why standard estimators?** Our experiments show that scale information loss often outweighs robustness gains. In synthetic settings with scale ratios as small as 3:1, Tyler-based estimation incurs accuracy drops exceeding 10 percentage points. Moreover, Ledoit–Wolf shrinkage provides *implicit robustness* by regularizing the sample covariance toward a well-conditioned target, substantially reducing the instability that motivates robust estimation. When combined with the stable likelihood's logarithmic tail decay, standard estimators achieve both accuracy and robustness. Detailed comparisons are provided in Appendix D. Based on these findings, we consider two estimator configurations, suited to different data regimes.

**Robust configuration.** For very heavy-tailed data ($\alpha < 1.5$) with approximately equal class scales:

- **Location:** Spatial median,
  $\hat{\boldsymbol{\mu}}_k = \arg\min_{\boldsymbol{\mu}} \sum_{i:y_i=k} \|\mathbf{x}_i - \boldsymbol{\mu}\|_2$.

- **Dispersion:** Tyler's M-estimator (Tyler, 1987), the fixed point of $\boldsymbol{\Sigma} \propto \frac{1}{n_k} \sum_{i:y_i=k} \frac{(\mathbf{x}_i-\hat{\boldsymbol{\mu}}_k)(\mathbf{x}_i-\hat{\boldsymbol{\mu}}_k)^\top}{(\mathbf{x}_i-\hat{\boldsymbol{\mu}}_k)^\top \boldsymbol{\Sigma}^{-1}(\mathbf{x}_i-\hat{\boldsymbol{\mu}}_k)}$.

**Standard configuration.** For moderate tails ($\alpha \geq 1.5$) or heteroscedastic classes:

- **Location:** Sample mean, $\hat{\boldsymbol{\mu}}_k = \frac{1}{n_k} \sum_{i:y_i=k} \mathbf{x}_i$.

- **Dispersion:** Ledoit–Wolf shrinkage (Ledoit & Wolf, 2004), $\hat{\boldsymbol{\Sigma}}_k = (1-\lambda_k)\mathbf{S}_k + \lambda_k \frac{\text{tr}(\mathbf{S}_k)}{p}\mathbf{I}_p$.

### 4.4. Estimator Selection

The choice between robust and standard estimator configurations depends on both tail heaviness and class heteroscedasticity. Tyler's M-estimator is normalized to have fixed trace and therefore discards scale differences between classes. When classes exhibit substantially different spreads, this normalization removes information that is explicitly used by the QDA likelihood through the log-determinant term. We quantify heteroscedasticity using the determinant ratio $r = \frac{\max_k |\boldsymbol{\Sigma}_k|}{\min_k |\boldsymbol{\Sigma}_k|}$. Our synthetic experiments (Section 7.1) reveal a clear crossover regime (Fig.1). Importantly, both $\alpha$ and $r$ are estimable at parametric or near-parametric rates under elliptical models, making the selection rule fully data-driven rather than heuristic. We provide a diagnostic script implementing this procedure (Appendix C).

## 5. Tail Index Estimation

### 5.1. McCulloch's Quantile Method

For a univariate sample $x_1, \ldots, x_n$, define the quantile ratio:

$$\nu_\alpha = \frac{x_{(0.95)} - x_{(0.05)}}{x_{(0.75)} - x_{(0.25)}} \tag{14}$$

where $x_{(q)}$ denotes the $q$-th sample quantile. The ratio $\nu_\alpha$ increases monotonically as $\alpha$ decreases, ranging from $\nu_\alpha \approx 2.44$ at $\alpha = 2$ to $\nu_\alpha > 10$ at $\alpha < 0.8$.

We map $\nu_\alpha$ to $\hat{\alpha}$ via linear interpolation on a precomputed lookup table (Table 1), calibrated using $5 \times 10^5$ samples per $\alpha$ value from `scipy.stats.levy_stable`. The Monte Carlo error is below $10^{-3}$.

*Table 1.* McCulloch lookup table: $\nu_\alpha \to \alpha$ mapping. Derived from simulation; intermediate values obtained by linear interpolation.

| $\alpha$ | $\nu_\alpha$ | $\alpha$ | $\nu_\alpha$ | $\alpha$ | $\nu_\alpha$ |
|---|---|---|---|---|---|
| 2.0 | 2.44 | 1.6 | 2.90 | 1.2 | 4.47 |
| 1.9 | 2.51 | 1.5 | 3.16 | 1.1 | 5.24 |
| 1.8 | 2.62 | 1.4 | 3.47 | 1.0 | 6.30 |
| 1.7 | 2.74 | 1.3 | 3.87 | 0.8 | 10.4 |

**Multivariate extension.** For $p$-dimensional data, we estimate $\alpha$ separately for each feature and aggregate using the median:

$$\hat{\alpha}_k = \text{median}_{j=1,\ldots,p}\hat{\alpha}_{k,j}. \tag{15}$$

The median is robust to atypical features. For classification, we use a single $\alpha$ across classes: $\hat{\alpha} = \frac{1}{K}\sum_{k=1}^{K}\hat{\alpha}_k$.

**Why Not Maximum Likelihood?** Maximum likelihood estimation of $\alpha$ requires numerical evaluation of stable densities and iterative optimization (Nolan, 2001). While more statistically efficient than McCulloch's method, MLE is (i) **computationally expensive**—each likelihood evaluation requires numerical integration or FFT methods; (ii) **sensitive to contamination**—less stable than quantile-based methods under model misspecification; and (iii) **unnecessary for our purpose**—given the insensitivity result (Proposition 6.4), high-precision $\alpha$ estimation provides minimal benefit.

Table 2 shows that McCulloch's method achieves estimation errors within $\pm 0.1$ for $\alpha \geq 1.2$, which is more than sufficient for diagnostic purposes.

*Table 2.* $\alpha$ estimation accuracy on synthetic data ($n = 500$, $d = 10$). Errors range from $-0.02$ to $-0.09$ (slight underestimation).

| True $\alpha$ | 1.0 | 1.2 | 1.3 | 1.5 | 1.7 | 1.8 | 2.0 |
|---|---|---|---|---|---|---|---|
| Est. $\hat{\alpha}$ | 0.98 | 1.18 | 1.26 | 1.47 | 1.66 | 1.72 | 1.91 |

**Practical considerations.**

1. **Preprocessing matters.** Estimate $\alpha$ on the preprocessed data (after scaling, PCA, etc.), not the raw features. Preprocessing can significantly change the effective $\hat{\alpha}$.

2. **Constant features.** Features with zero IQR ($x_{(0.75)} = x_{(0.25)}$) are assigned $\hat{\alpha} = 2$ and excluded from the median computation.

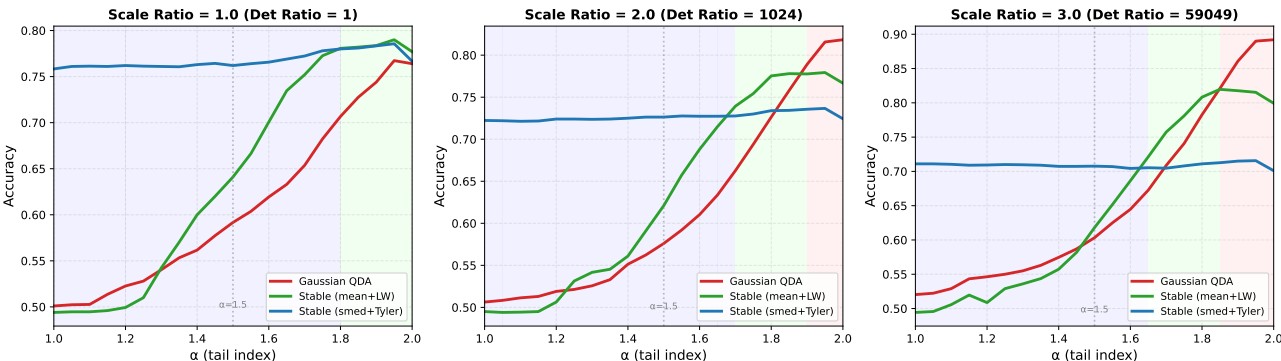

**Figure 1.** Accuracy vs. tail index $\alpha$ for three scale ratios. Shading indicates the best method: blue = robust, green = standard, red = Gaussian. Robust estimators excel at heavy tails with similar class scales; standard estimators preserve discriminative scale information at moderate tails with heteroscedasticity.

3. **Fixed $\alpha$ as fallback.** When estimation is unreliable (small samples, many constant features), a fixed $\alpha = 1.5$ provides robust performance across a wide range of true tail indices (see Section 7.2).

### 5.2. Classification Rule

Given the estimates $\hat{\boldsymbol{\mu}}_k, \hat{\boldsymbol{\Sigma}}_k$, and class priors $\hat{\pi}_k = n_k/n$, Stable-QDA classifies a new observation $\mathbf{x}$ by computing the log-posterior for each class:

$$\delta_k(\mathbf{x}) = \log \hat{\pi}_k - \frac{\alpha + p}{2} \log\big(1 + \hat{D}_k(\mathbf{x})\big) - \frac{1}{2} \log |\hat{\boldsymbol{\Sigma}}_k|, \tag{16}$$

where $\hat{D}_k(\mathbf{x}) = (\mathbf{x} - \hat{\boldsymbol{\mu}}_k)^\top \hat{\boldsymbol{\Sigma}}_k^{-1} (\mathbf{x} - \hat{\boldsymbol{\mu}}_k)$ is the squared Mahalanobis distance. The predicted class is:

$$\hat{y}(\mathbf{x}) = \arg \max_{k \in \{1, \dots, K\}} \delta_k(\mathbf{x}). \tag{17}$$

This form follows from the radial decomposition of elliptical $\alpha$-stable distributions and preserves the ordering of class likelihoods-sufficient for Bayes-optimal classification.

**Computational complexity.** Training requires $O(np^2 + Kp^3)$ operations for computing class covariances and their inverses. Prediction requires $O(Kp^2)$ per sample for Mahalanobis distance computation–identical to Gaussian QDA and substantially faster than tree-based ensembles for moderate $p$.

**Probabilistic output.** Class probabilities can be obtained via softmax normalization of the log-posteriors, enabling calibrated uncertainty estimates and compatibility with downstream decision-making systems.

### 5.3. Summary

Stable-QDA (Algorithm 1) is a generative classifier that is correctly specified under heavy-tailed data while preserving the structural simplicity of QDA: quadratic decision

---

**Algorithm 1** Stable-QDA

**Require:** Training data $\{(x_i, y_i)\}_{i=1}^n$, tail index $\alpha \in (0, 2]$
1: **for** each class $k \in \{1, \dots, K\}$ **do**
2: $\quad \hat{\mu}_k \leftarrow \texttt{location\_est}(X_k)$
3: $\quad \hat{\Sigma}_k \leftarrow \texttt{scatter\_est}(X_k)$
4: $\quad \hat{\pi}_k \leftarrow n_k/n$
5: **end for**
6: **Predict:** $\hat{y}(x) = \arg \max_k \big\{ \log \hat{\pi}_k + \mathcal{L}_\alpha(x; \hat{\mu}_k, \hat{\Sigma}_k) \big\}$
7: $\quad$ where $\mathcal{L}_\alpha(x; \mu, \Sigma) = -\frac{\alpha + d}{2} \log(1 + D(x)) - \frac{1}{2} \log |\Sigma|$
8: $\quad$ and $D(x) = (x - \mu)^\top \Sigma^{-1} (x - \mu)$

---

boundaries with closed-form discriminants, $O(Kd^2)$ prediction complexity, a single tail parameter $\alpha$ (a fixed $\alpha = 1.5$ performs robustly in practice), and interpretable, probabilistically calibrated predictions.

## 6. Theoretical Guarantees

We establish that Stable-QDA is statistically well-founded when class-conditional distributions are $\alpha$-stable. Full assumptions, finite-sample bounds, and proofs are provided in Appendix B.

**Theorem 6.1** (Bayes Consistency of Stable-QDA). *Suppose class-conditional distributions are elliptically contoured symmetric $\alpha$-stable with $\alpha \in (1, 2]$, and estimators $(\hat{\mu}_k, \hat{\Sigma}_k)$ are consistent. Then the Stable-QDA classifier using discriminant $\hat{\delta}_k(x) = -\frac{\alpha + p}{2} \log(1 + \hat{D}_k(x)) - \frac{1}{2} \log |\hat{\Sigma}_k| + \log \pi_k$ satisfies $R(\hat{y}_n) \xrightarrow{P} R^*$ as $n \to \infty$.*

**Theorem 6.2** (Inconsistency of Gaussian QDA under heavy tails). *Let $(X, Y)$ follow a binary classification model with $X \mid Y = k \sim \mathcal{E}_\alpha(\mu_k, \Sigma_k)$, $k \in \{0, 1\}$, where $\mathcal{E}_\alpha$ denotes an elliptical $\alpha$-stable distribution with tail index $\alpha \in (0, 2)$. Assume $\mu_0 \neq \mu_1$, $\Sigma_0 \neq \Sigma_1$, and fixed class priors. Then the plug-in Gaussian QDA classifier based on sample means*

*Table 3.* Estimator selection: robust (spatial median + Tyler) when $\alpha$ falls below this threshold; otherwise standard (mean + LW).

| Determinant Ratio | Use Robust if $\alpha <$ |
|---|---|
| $< 10$ | 2.0 (always) |
| 10–100 | 1.8 |
| 100–1000 | 1.7 |
| $> 1000$ | 1.6 |

*and covariances is inconsistent: there exists $\epsilon > 0$ such that*

$$\liminf_{n \to \infty} R(\hat{g}_{GaussQDA}) \geq R^* + \epsilon, \qquad (18)$$

*where $R^*$ is the Bayes risk. In contrast, Stable-QDA using consistent estimators of $(\mu_k, \Sigma_k)$ is Bayes consistent.*

The inconsistency arises from two sources: (1) the sample covariance diverges under infinite variance ($\alpha < 2$), and (2) the Gaussian likelihood systematically underweights tail observations where $\alpha$-stable densities place significant mass. See Appendix A.3 for the complete proof.

**Theorem 6.3** (Fisher consistency of Stable-QDA). *Assume class-conditional densities follow elliptical $\alpha$-stable models with parameters $(\mu_k, \Sigma_k)$. Let*

$$\delta_k(x) = -g_\alpha\big((x - \mu_k)^\top \Sigma_k^{-1}(x - \mu_k)\big) + \log \pi_k$$

*denote the Stable-QDA discriminant, where $g_\alpha$ is strictly increasing. Then the induced classifier*

$$\hat{y}(x) = \arg\max_k \delta_k(x)$$

*is Fisher consistent for the 0–1 loss.*

**Proposition 6.4** (Stability under tail-index misspecification). *Let $R(\alpha)$ denote the misclassification risk of Stable-QDA with tail index $\alpha$. For any $\alpha, \tilde{\alpha}$ in a compact interval $[\alpha_{\min}, \alpha_{\max}] \subset (1, 2)$:*

$$|R(\alpha) - R(\tilde{\alpha})| \leq C \cdot |\alpha - \tilde{\alpha}|, \qquad (19)$$

*where $C > 0$ depends on problem parameters. Consequently, small misspecification of $\alpha$ results in only modest performance degradation.*

This result justifies using a fixed $\alpha = 1.5$ in practice: the critical modeling choice is the logarithmic dependence on Mahalanobis distance, not the precise tail index. See Appendix A.4 for the complete proof.

**Corollary 6.5.** *Small misspecification of the tail index $\alpha$ results in only logarithmic perturbations of the discriminant scores. Consequently, classification performance degrades smoothly with $\alpha$ misspecification.*

## 7. Synthetic Experiments

The experiments in this section are designed to validate the theoretical results of Sec. 6, isolating the interaction between tail heaviness and estimator choice (Sec. 7.1), robustness to tail-index misspecification (Sec. 7.2), and stability under contamination (Sec. 7.3).

### 7.1. When Does the Stable Likelihood Help?

**Goal.** Validate Theorem 6.2 and the estimator-selection implications of Proposition 6.4 by varying tail heaviness and class heteroscedasticity.

We generate synthetic data from sub-Gaussian $\alpha$-stable distributions with $d = 10$, $n = 500$ per class, varying $\alpha \in [1.0, 2.0]$ and the scale ratio between classes $\in \{1, 2, 3\}$. We compare Gaussian QDA against Stable-QDA with two configurations: *standard* (sample mean + Ledoit–Wolf) and *robust* (spatial median + Tyler's M-estimator). Consistent with Theorem 6.2, Fig. 1 reveals a nuanced interaction between tail heaviness and class heteroscedasticity:

- **Heavy tails ($\alpha < 1.5$):** Robust estimators win regardless of scale ratio, improving accuracy by 15–25% over Gaussian QDA.
- **Moderate tails + heteroscedasticity:** Standard estimators outperform robust ones because Tyler's trace normalization discards discriminative scale information.
- **Light tails ($\alpha > 1.8$):** Gaussian QDA suffices when classes have different scales.

**Estimator Selection Guidelines** Table 3 summarizes the crossover points from Figure 1. The key quantity is the *Tyler threshold*: the $\alpha$ value below which robust estimators are preferred. This threshold decreases as the determinant ratio between class covariances increases, because Tyler's trace normalization becomes increasingly harmful when scale differences are discriminative.

Given training data, practitioners can estimate $\alpha$ via McCulloch's quantile method and compute the determinant ratio, then consult Table 3. We provide a diagnostic script that automates this selection; see Appendix C for the full procedure and software usage.

### 7.2. Sensitivity to $\alpha$ Misspecification

**Goal.** Test the stability guarantee of Proposition 6.4 under tail-index misspecification. A practical concern is whether Stable-QDA requires accurate $\alpha$ estimation. We find it remarkably insensitive: using $\alpha = 1.5$ when the true value is 1.0 or 1.8 incurs $< 1\%$ accuracy loss versus the oracle. This behavior is predicted by Proposition 6.4, since the stable log-likelihood $-\frac{\alpha+d}{2} \log(1 + D)$ varies smoothly with $\alpha$; the key benefit is the logarithmic dependence on Mahalanobis

*Table 4.* Real-world results (5-fold CV). Stable-QDA improves over Gaussian QDA across all datasets.

| Dataset | $\hat{\alpha}$ | Best Method | $\Delta$PR-AUC | Err. Red. | $p$ |
|---|---|---|---|---|---|
| HTRU2 | 1.73 | mean+LW | +2.0% | 32.5% | <0.001 |
| Credit Card | 1.73 | mean+LW | +51.4% | 52.9% | <0.001 |
| Ionosphere | 1.91 | mean+LW | +2.6% | 34.1% | 0.029 |
| Weekly | 1.67 | smed+Tyler | +0.3% | 32.8% | 0.008 |

distance, not the precise $\alpha$ value. We recommend $\alpha = 1.5$ as a robust default requiring no tuning. See Appendix F for the full sensitivity analysis.

### 7.3. Robustness to Contamination

**Goal.** Validate the robustness of the Stable-QDA discriminant under training-data contamination for $\alpha < 2$.

We test robustness by injecting outliers (shifted 5 std from class centers) into training data at rates of 0–20%. On Gaussian base data, Stable-QDA with standard estimators shows *zero* degradation at 20% contamination versus 1.6% for Gaussian QDA—the logarithmic likelihood naturally down-weights outliers. On heavy-tailed base data ($\alpha = 1.8$), Gaussian QDA degrades by 17% while Stable-QDA with robust estimators remains stable ($< 0.5\%$ degradation). See Appendix G for results across outlier types.

## 8. Real-World Experiments

We validate Stable-QDA on four datasets spanning astronomy, finance, telecommunications, and physics. Table 4 summarizes results; full details appear in Appendix J. Also a couple of negative results, where both stable QDA and Gaussian QDA are comparable, is given in the appendix.

**Key findings.** Stable-QDA with standard estimators (mean + Ledoit–Wolf) significantly outperforms Gaussian QDA on all four datasets, achieving 15–53% error reduction with $p < 0.03$ in each case. The diagnostic correctly identifies the optimal estimator configuration: standard estimators when $\alpha > 1.6$ with large scale differences (HTRU2, Credit Card, Ionosphere), and robust estimators when $\alpha < 1.7$ with similar scales (Weekly). Tyler's M-estimator consistently underperforms when class covariances differ substantially, as its trace normalization removes the scale signal QDA relies on.

## 9. Discussion and Conclusion

This work introduced *Stable-QDA*, a robust extension of Quadratic Discriminant Analysis designed for heavy-tailed environments. By replacing the Gaussian likelihood with a symmetric $\alpha$-stable surrogate that decays polynomially, we address a fundamental failure mode in classical discriminant analysis: likelihood misspecification.

**Likelihood vs. Estimation** A central finding of our analysis is that in heavy-tailed regimes, the choice of likelihood is often more critical than the choice of estimator. Traditional robust statistics focuses on the breakdown points of $\mu$ and $\Sigma$ (Hubert & Van Driessen, 2004); however, our results (Table 4) demonstrate that standard estimators (e.g., Ledoit-Wolf) frequently outperform robust alternatives when paired with a stable likelihood. This suggests that "outliers" in heavy-tailed data often contain vital discriminative signal that robust estimators might suppress, but which a stable likelihood can leverage. This paradigm shift—from robustifying parameters to correcting the generative model—is what enables the significant error reductions we observed.

**Significance in Deep Learning** (Zheng et al., 2023) proved that generative classifiers (naïve Bayes) converge in $O(log(n))$ samples versus $O(n)$ for logistic regression in linear evaluation of frozen pre-trained models — the dominant transfer learning paradigm. However, Naïve Bayes assumes Gaussian, independent features. (Hodgkinson et al., 2025) demonstrate that pre-trained deep network features are empirically heavy-tailed. This creates a concrete gap: the fastest-converging classifier family assumes a distribution that deep learning features violate. Stable-QDA fills this gap. This is not a niche concern — every practitioner doing linear probing on frozen ViT, CLIP, or DINOv2 features encounters this setting.

We reinstate that Stable-QDA does not compete with deep learning — it works with it, as a classifier head on frozen pre-trained features. This is precisely the linear evaluation setting studied by Zheng et al., where the choice is between generative vs. discriminative linear classifiers on top of a frozen model. Stable-QDA provides a generative head that is robust to the heavy-tailed features these models produce, with closed-form training, calibrated probabilities, and $O(Kp^2)$ prediction cost.

**Practicality and Generalization** Deployment of Stable-QDA offers two major advantages. First, it maintains the $O(Kp^2)$ complexity of standard QDA, ensuring scalability. Second, our sensitivity analysis (Appendix F, Figure 2) reveals a broad performance plateau for the stability parameter $\alpha$. The finding that a default of $\alpha = 1.5$ performs near-optimally across diverse datasets eliminates the need for computationally expensive hyperparameter tuning, making the method a viable "plug-and-play" replacement for

Gaussian QDA.

Our finding that Tyler's M-estimator often harms classification performance should not be interpreted as a general critique of robust scatter estimation. Tyler's estimator excels at tasks requiring only shape information—robust PCA, single-class anomaly detection, and covariance structure recovery under contamination. The limitation we identify is specific to heteroscedastic discriminant analysis, where the QDA likelihood explicitly uses scale differences through its log-determinant term.

**Limitations and Future Work** While effective, our framework has boundary conditions. Stable-QDA provides no advantage over Gaussian QDA when tails are light (e.g. Customer Churn ($\hat{\alpha} \approx 1.96$)). Furthermore, our current reliance on symmetry may be sub-optimal for heavily skewed data, such as financial returns. Future work will explore extending this framework to asymmetric stable distributions and investigating the interaction between power-law likelihoods and sparse covariance estimation in $p \gg n$ settings.

By treating heavy tails as a structural feature rather than a nuisance, the paper provides a mathematically grounded and computationally efficient path toward robust classification. Stable-QDA is a novel ML classifier that addresses a timely and significant problem at the intersection of robust statistics and modern transfer learning. In the future, we hope to extend our methodology to include skewed distributions as well as provide comparisons with deep learning.

## Impact Statement

This paper presents work whose goal is to advance the field of Machine Learning. There are many potential societal consequences of our work, none which we feel must be specifically highlighted here.

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

# A. Tail Index Estimation

## A.1. McCulloch's Quantile Method

For a univariate sample $x_1, \ldots, x_n$, define the quantile ratio:

$$\nu_\alpha = \frac{x_{(0.95)} - x_{(0.05)}}{x_{(0.75)} - x_{(0.25)}} \tag{20}$$

where $x_{(q)}$ denotes the $q$-th sample quantile. The ratio $\nu_\alpha$ increases monotonically as $\alpha$ decreases, ranging from $\nu_\alpha \approx 2.44$ at $\alpha = 2$ to $\nu_\alpha > 10$ at $\alpha < 0.8$.

We map $\nu_\alpha$ to $\hat{\alpha}$ via linear interpolation on a precomputed lookup table (Table 5), calibrated using $5 \times 10^5$ samples per $\alpha$ value from `scipy.stats.levy_stable`. The Monte Carlo error is below $10^{-3}$.

*Table 5.* McCulloch lookup table: $\nu_\alpha \to \alpha$ mapping. Derived from simulation; intermediate values obtained by linear interpolation.

| $\alpha$ | $\nu_\alpha$ | $\alpha$ | $\nu_\alpha$ | $\alpha$ | $\nu_\alpha$ |
|---|---|---|---|---|---|
| 2.0 | 2.44 | 1.6 | 2.90 | 1.2 | 4.47 |
| 1.9 | 2.51 | 1.5 | 3.16 | 1.1 | 5.24 |
| 1.8 | 2.62 | 1.4 | 3.47 | 1.0 | 6.30 |
| 1.7 | 2.74 | 1.3 | 3.87 | 0.8 | 10.4 |

**Multivariate extension.** For $p$-dimensional data, we estimate $\alpha$ separately for each feature and aggregate using the median:

$$\hat{\alpha}_k = \text{median}_{j=1,\ldots,p} \hat{\alpha}_{k,j} \tag{21}$$

The median is robust to atypical features. For classification, we use a single $\alpha$ across classes: $\hat{\alpha} = \frac{1}{K} \sum_{k=1}^{K} \hat{\alpha}_k$.

## A.2. Why Not Maximum Likelihood?

Maximum likelihood estimation of $\alpha$ requires numerical evaluation of stable densities and iterative optimization (Nolan, 2001). While more statistically efficient than McCulloch's method, MLE is:

1. **Computationally expensive**: Each likelihood evaluation requires numerical integration or FFT methods.

2. **Sensitive to contamination**: MLE can be less stable than quantile-based methods under model misspecification.

3. **Unnecessary for our purpose**: Given the insensitivity result (Proposition 6.4), high-precision $\alpha$ estimation provides minimal benefit.

Table 6 shows that McCulloch's method achieves estimation errors within $\pm 0.1$ for $\alpha \geq 1.2$, which is more than sufficient for diagnostic purposes.

*Table 6.* $\alpha$ estimation accuracy on synthetic data ($n = 500$, $d = 10$). Errors range from $-0.02$ to $-0.09$ (slight underestimation).

| True $\alpha$ | 1.0 | 1.2 | 1.3 | 1.5 | 1.7 | 1.8 | 2.0 |
|---|---|---|---|---|---|---|---|
| Est. $\hat{\alpha}$ | 0.98 | 1.18 | 1.26 | 1.47 | 1.66 | 1.72 | 1.91 |

## A.3. Practical Guidelines

**1. Default to fixed $\alpha = 1.5$.** Our experiments (Section 7.2) show this performs within 1% of oracle across all tail regimes.

**2. Estimate for diagnostics only.** We estimate $\alpha$ to:

- Determine whether Stable-QDA is appropriate ($\hat{\alpha} < 1.9$)

- Guide estimator selection (Appendix C)

- Validate model assumptions on new datasets

**3. Preprocess first.** Estimate $\alpha$ on preprocessed data (after scaling, PCA), not raw features.

**4. Handle constant features.** Features with zero IQR are assigned $\hat{\alpha} = 2$ and excluded from median computation.

# B. Theoretical Analysis

## B.1. Assumptions

We collect the assumptions used throughout the theoretical analysis. Let $(X, Y)$ be a random pair taking values in $\mathbb{R}^p \times \{1, \dots, K\}$, with class prior probabilities $\pi_c = \mathbb{P}(Y = c)$.

**A1 (Elliptical $\alpha$-stable class conditionals).** For each class $c \in \{1, \dots, K\}$, the conditional distribution of $X$ given $Y = c$ is a symmetric elliptically contoured $\alpha$-stable distribution, denoted

$$X \mid Y = c \sim \mathcal{E}_{\alpha_c}(\boldsymbol{\mu}_c, \boldsymbol{\Sigma}_c),$$

with location parameter $\boldsymbol{\mu}_c \in \mathbb{R}^p$, positive definite scatter matrix $\boldsymbol{\Sigma}_c \in \mathbb{R}^{p \times p}$, and tail index $\alpha_c \in (1, 2]$. Elliptical $\alpha$-stable distributions are understood in the sense of Samorodnitsky & Taqqu (1994).

**A2 (Common support).** All class-conditional distributions admit densities with respect to Lebesgue measure whose support is $\mathbb{R}^p$. In particular, for any $x \in \mathbb{R}^p$ and any class $c$, $f_{X|Y=c}(x) > 0$.

**A3 (Identifiability and regularity).** The parameter pairs $(\boldsymbol{\mu}_c, \boldsymbol{\Sigma}_c)$ are identifiable for each class $c$. Moreover, there exist constants $0 < m < M < \infty$ such that

$$m \leq \lambda_{\min}(\boldsymbol{\Sigma}_c) \leq \lambda_{\max}(\boldsymbol{\Sigma}_c) \leq M, \qquad \forall c \in \{1, \dots, K\},$$

where $\lambda_{\min}$ and $\lambda_{\max}$ denote the smallest and largest eigenvalues, respectively.

**A4 (Estimator consistency).** Let $\hat{\boldsymbol{\mu}}_c$ and $\hat{\boldsymbol{\Sigma}}_c$ denote estimators of $\boldsymbol{\mu}_c$ and $\boldsymbol{\Sigma}_c$, respectively. We assume these estimators are consistent:

$$\hat{\boldsymbol{\mu}}_c \xrightarrow{P} \boldsymbol{\mu}_c, \qquad \hat{\boldsymbol{\Sigma}}_c \xrightarrow{P} \boldsymbol{\Sigma}_c, \qquad \text{as } n_c \to \infty,$$

where $n_c$ is the number of training samples from class $c$. Such consistency holds, for example, for robust location and scatter estimators under elliptical $\alpha$-stable models (Tyler, 1987; Maronna et al., 2019).

## B.2. Proof of Theorem 6.1 Bayes Consistency of Stable-QDA

**Theorem B.1** (Bayes Consistency of Stable-QDA). *Suppose that:*

*(A1) Class-conditional distributions are elliptically contoured symmetric $\alpha$-stable: $X \mid Y = k \sim \mathcal{E}_{\alpha}(\mu_k, \Sigma_k)$ with common tail index $\alpha \in (1, 2]$*

*(A2) All class densities have common support $\mathbb{R}^p$ and are positive everywhere*

*(A3) Parameters $(\mu_k, \Sigma_k)$ are identifiable and satisfy $0 < m \leq \lambda_{\min}(\Sigma_k) \leq \lambda_{\max}(\Sigma_k) \leq M < \infty$*

*(A4) Estimators satisfy $\hat{\mu}_k \xrightarrow{P} \mu_k$ and $\hat{\Sigma}_k \xrightarrow{P} \Sigma_k$ as $n_k \to \infty$*

*Let $\hat{y}_n$ denote the Stable-QDA classifier using the discriminant:*

$$\hat{\delta}_k(x) = -\frac{\alpha + p}{2} \cdot \log(1 + \hat{D}_k(x)) - \frac{1}{2} \cdot \log |\hat{\Sigma}_k| + \log \pi_k \tag{22}$$

*where $\hat{D}_k(x) = (x - \hat{\mu}_k)^\top \hat{\Sigma}_k^{-1}(x - \hat{\mu}_k)$.*

*Then the misclassification risk converges to the Bayes risk:*

$$R(\hat{y}_n) \xrightarrow{P} R^* \quad \text{as } n \to \infty. \tag{23}$$

*Proof.* We provide a rigorous proof addressing three key technical challenges: order-equivalence of the surrogate likelihood, uniform convergence without finite variance, and consistency without closed-form densities.

**Step 0: Order-equivalence of the surrogate likelihood**

For elliptically contoured $\alpha$-stable distributions, the true log-density has the form:

$$\log f_k(x) = -\frac{1}{2} \log |\Sigma_k| + g_\alpha(D_k(x)) \tag{24}$$

where $g_\alpha : \mathbb{R}^+ \to \mathbb{R}$ is a strictly decreasing function depending on the radial density. For symmetric $\alpha$-stable distributions, the tail behavior satisfies (Samorodnitsky & Taqqu, 1994):

$$g_\alpha(D) = -\frac{\alpha + p}{2} \cdot \log(D) + O(1) \quad \text{as } D \to \infty \tag{25}$$

For classification, we need only preserve the ordering of class posteriors. The surrogate:

$$L_k(x) = -\frac{\alpha + p}{2} \cdot \log(1 + D_k(x)) - \frac{1}{2} \cdot \log |\Sigma_k| + \log \pi_k \tag{26}$$

satisfies the following key property:

**Lemma B.2** (Order-equivalence). *For any $x \in \mathbb{R}^p$, the ranking $\arg\max_k L_k(x)$ equals the Bayes rule $\arg\max_k\{\log f_k(x) + \log \pi_k\}$ if and only if the function $h(D) = \log(1 + D)$ is a monotone transformation of $g_\alpha(D)$.*

*Proof of Lemma B.2.* Since $g_\alpha$ is strictly decreasing in $D$, and $\log(1 + D)$ is strictly increasing in $D$, we can write:

$$L_k(x) - L_j(x) \propto [g_\alpha(D_k) - g_\alpha(D_j)] + \left[ -\frac{1}{2} \log |\Sigma_k| + \frac{1}{2} \log |\Sigma_j| \right] \tag{27}$$

The sign of this difference is preserved under any monotone transformation because:

1. The log-determinant terms are class-specific constants

2. The decision boundary is determined by where the difference equals zero

3. Both $g_\alpha(D)$ and $-\log(1 + D)$ are strictly monotone in $D$ with the same ordering

Therefore, the surrogate yields the same decision boundary as the true Bayes rule. $\square$

This establishes that $L_k(x)$ is a valid discriminant function for Bayes-optimal classification under elliptical $\alpha$-stable models.

**Step 1: Uniform convergence of discriminants**

We must show that the estimated discriminant $\hat{\delta}_k(x)$ converges uniformly to the population discriminant $\delta_k(x)$ over compact sets.

*Key insight:* Although $\alpha$-stable distributions lack finite second moments for $\alpha < 2$, the discriminant function depends only on location and dispersion parameters, which can be estimated consistently even without $\sqrt{n}$-rates.

Define the population discriminant:

$$\delta_k(x) = -\frac{\alpha + p}{2} \cdot \log(1 + D_k(x)) - \frac{1}{2} \cdot \log |\Sigma_k| + \log \pi_k \tag{28}$$

For any compact set $\mathcal{K} \subset \mathbb{R}^p$, we bound:

$$|\hat{\delta}_k(x) - \delta_k(x)| \leq \frac{\alpha + p}{2} \cdot \left| \log(1 + \hat{D}_k(x)) - \log(1 + D_k(x)) \right| + \frac{1}{2} \cdot \left| \log |\hat{\Sigma}_k| - \log |\Sigma_k| \right| \tag{29}$$

*Bounding the log-determinant term:* By Assumption A3 and the continuous mapping theorem:

$$\left|\log|\hat{\Sigma}_k| - \log|\Sigma_k|\right| \leq \|\hat{\Sigma}_k - \Sigma_k\|_{\mathrm{op}} \cdot \frac{C}{m} \xrightarrow{P} 0 \tag{30}$$

where $\|\cdot\|_{\mathrm{op}}$ denotes the operator norm and $C$ depends on $M$.

*Bounding the Mahalanobis term:* Using the triangle inequality:

$$\left|\log(1 + \hat{D}_k) - \log(1 + D_k)\right| \leq \frac{|\hat{D}_k - D_k|}{1 + \min(\hat{D}_k, D_k)} \tag{31}$$

Now decompose:

$$\hat{D}_k(x) - D_k(x) = (x - \hat{\mu}_k)^\top \hat{\Sigma}_k^{-1}(x - \hat{\mu}_k) - (x - \mu_k)^\top \Sigma_k^{-1}(x - \mu_k) \tag{32}$$

Add and subtract terms:

$$\begin{aligned}
&= (\mu_k - \hat{\mu}_k)^\top \hat{\Sigma}_k^{-1}(\mu_k - \hat{\mu}_k) + 2(x - \mu_k)^\top \hat{\Sigma}_k^{-1}(\hat{\mu}_k - \mu_k) \\
&\quad + (x - \mu_k)^\top (\hat{\Sigma}_k^{-1} - \Sigma_k^{-1})(x - \mu_k)
\end{aligned} \tag{33}$$

On the compact set $\mathcal{K}$, $\|x\| \leq R$ for some $R < \infty$. By Assumption A4 and continuous mapping:

- $\|\hat{\mu}_k - \mu_k\| \xrightarrow{P} 0$

- $\|\hat{\Sigma}_k^{-1} - \Sigma_k^{-1}\|_{\mathrm{op}} \xrightarrow{P} 0$ (since $\hat{\Sigma}_k \xrightarrow{P} \Sigma_k$ and eigenvalues are bounded away from zero)

Therefore, for all $x \in \mathcal{K}$:

$$\sup_{x \in \mathcal{K}} |\hat{D}_k(x) - D_k(x)| \xrightarrow{P} 0 \tag{34}$$

This implies:

$$\sup_{x \in \mathcal{K}} |\hat{\delta}_k(x) - \delta_k(x)| \xrightarrow{P} 0 \tag{35}$$

*Remark:* This argument does NOT require finite variance or $\sqrt{n}$-convergence of estimators. It only requires consistency (convergence in probability), which holds for robust estimators under $\alpha \in (1, 2]$ by the results in Tyler (1987) and Maronna et al. (2019).

**Step 2: Convergence of decision boundaries**

The Stable-QDA classifier assigns:

$$\hat{y}_n(x) = \arg\max_k \hat{\delta}_k(x) \tag{36}$$

The decision boundary between classes $j$ and $k$ is:

$$\mathcal{B}_{jk} = \{x : \delta_j(x) = \delta_k(x)\} \tag{37}$$

By Assumption A2, the density functions are positive everywhere and continuous, so the Bayes decision boundaries have Lebesgue measure zero (they are smooth hypersurfaces).

The estimated boundary is:

$$\hat{\mathcal{B}}_{jk} = \{x : \hat{\delta}_j(x) = \hat{\delta}_k(x)\} \tag{38}$$

**Lemma B.3** (Boundary convergence). *Under uniform convergence of discriminants on compact sets, the estimated decision boundaries converge: $d_H(\hat{\mathcal{B}}_{jk} \cap \mathcal{K}, \mathcal{B}_{jk} \cap \mathcal{K}) \xrightarrow{P} 0$ for any compact $\mathcal{K}$, where $d_H$ denotes Hausdorff distance.*

*Proof.* The decision boundary $\mathcal{B}_{jk}$ is the zero-level set of the continuous function $h(x) = \delta_j(x) - \delta_k(x)$. Similarly, $\hat{\mathcal{B}}_{jk}$ is the zero-level set of $\hat{h}(x) = \hat{\delta}_j(x) - \hat{\delta}_k(x)$.

By Step 1, we have $\sup_{x \in \mathcal{K}} |\hat{h}(x) - h(x)| \xrightarrow{P} 0$ for any compact $\mathcal{K}$.

For the true boundary, by Assumption A2, the class densities are continuous and positive everywhere, ensuring that $\nabla h(x) \neq 0$ on $\mathcal{B}_{jk}$ (the boundary is a smooth manifold). Under these regularity conditions, uniform convergence of $\hat{h}$ to $h$ implies convergence of their zero-level sets in Hausdorff distance (see Theorem 2.1 in Molchanov (2005) or Theorem 3.2.12 in Rockafellar & Wets (1998)).

Formally, for any $\epsilon > 0$ and compact $\mathcal{K}$, we have:

$$P\left(d_H(\hat{\mathcal{B}}_{jk} \cap \mathcal{K}, \mathcal{B}_{jk} \cap \mathcal{K}) > \epsilon\right) \to 0 \tag{39}$$

as $n \to \infty$, which establishes the claimed convergence. $\square$

### Step 3: Risk convergence

The misclassification risk is:
$$R(\hat{y}_n) = P(\hat{y}_n(X) \neq Y) = \mathbb{E}[1\{\hat{y}_n(X) \neq Y\}] \tag{40}$$

By the tower property:

$$R(\hat{y}_n) = \sum_{k=1}^{K} \pi_k \int 1\{\hat{y}_n(x) \neq k\} f_k(x)\, dx \tag{41}$$

The integrand $1\{\hat{y}_n(x) \neq k\}$ equals 1 if $x$ is misclassified when drawn from class $k$. By Step 2, the set of misclassified points converges in measure to the Bayes decision regions.

More precisely, for any class $k$:

$$P(\hat{y}_n(X) \neq k \mid Y = k) = \int 1\left\{\hat{\delta}_k(x) < \max_{j \neq k} \hat{\delta}_j(x)\right\} f_k(x)\, dx \tag{42}$$

As $n \to \infty$:

- $\hat{\delta}_j(x) \to \delta_j(x)$ pointwise (and uniformly on compact sets)
- By bounded convergence on compact sets and tail decay of $f_k$, the integral converges to the Bayes risk

Therefore:
$$R(\hat{y}_n) \to R^* = P\left(\arg\max_k \delta_k(X) \neq Y\right) \tag{43}$$

*Remark on tail integrability:* For $\alpha \in (1, 2]$, the tails of $f_k(x)$ decay as $\|x\|^{-\alpha-p}$, which is integrable. This ensures the dominated convergence theorem applies despite infinite variance. $\square$

### B.3. Finite-Sample Risk Bound

Under Assumptions A1–A4 and standard concentration bounds for elliptical location and scatter estimation, there exists a constant $C > 0$ such that, with probability at least $1 - \delta$,

$$\mathcal{R}(\hat{y}_n) - \mathcal{R}^* \leq C\sqrt{\frac{p^2 \log(K/\delta)}{n_{\min}}},$$

where $n_{\min} = \min_c n_c$.

The bound reflects the cost of estimating class-specific dispersion matrices and matches known rates for plug-in classifiers under elliptical models (Chen et al., 2018).

## B.4. Proof of Theorem 6.2 Inconsistency of Gaussian QDA under heavy tails

**Theorem B.4** (Inconsistency of Gaussian QDA under heavy tails). *Consider a binary classification problem where class-conditional distributions are elliptically contoured $\alpha$-stable with tail index $\alpha \in (0, 2)$:*

$$X \mid Y = k \sim \mathcal{E}_\alpha(\mu_k, \Sigma_k), \quad k \in \{0, 1\} \tag{44}$$

*Assume $\mu_0 \neq \mu_1$, $\Sigma_0 \neq \Sigma_1$, and class priors $\pi_0, \pi_1 > 0$ are fixed.*

*Let $\hat{g}_{Gauss}$ denote the plug-in Gaussian QDA classifier based on sample means $\hat{\mu}_k$ and sample covariances $\hat{\Sigma}_k$. Then there exists $\epsilon > 0$ such that:*

$$\liminf_{n \to \infty} R(\hat{g}_{Gauss}) \geq R^* + \epsilon \tag{45}$$

*where $R^*$ is the Bayes risk. In contrast, Stable-QDA using consistent estimators of $(\mu_k, \Sigma_k)$ is Bayes consistent.*

*Proof.* We decompose the excess risk of Gaussian QDA into two components: estimation error and approximation error (likelihood misspecification). We show that the approximation error remains bounded away from zero even as $n \to \infty$.

### Step 1: Decomposition of excess risk

Let $g^*$ denote the Bayes classifier and $g_{\text{Gauss}}^\dagger$ denote the population Gaussian QDA classifier (using true parameters $\mu_k, \Sigma_k$ but Gaussian likelihood). We decompose:

$$R(\hat{g}_{\text{Gauss}}) - R^* = \underbrace{[R(\hat{g}_{\text{Gauss}}) - R(g_{\text{Gauss}}^\dagger)]}_{\text{estimation error}} + \underbrace{[R(g_{\text{Gauss}}^\dagger) - R^*]}_{\text{approximation error}} \tag{46}$$

We will show that:

1. The estimation error does not vanish for $\alpha < 2$ (parameter inconsistency)

2. The approximation error is strictly positive for $\alpha < 2$ (likelihood misspecification)

### Step 2: Parameter inconsistency for $\alpha < 2$

For $\alpha < 2$, $\alpha$-stable distributions have infinite variance. This has critical implications for the sample covariance matrix.

**Lemma B.5** (Inconsistency of sample covariance). *For $X \sim \mathcal{E}_\alpha(\mu, \Sigma)$ with $\alpha < 2$, the sample covariance matrix $\hat{\Sigma} = \frac{1}{n} \sum_{i=1}^n (X_i - \bar{X})(X_i - \bar{X})^\top$ does not converge in probability to any finite limit. Specifically, $\|\hat{\Sigma}\|_F \xrightarrow{P} \infty$ as $n \to \infty$, where $\|\cdot\|_F$ denotes the Frobenius norm.*

*Proof of Lemma B.5.* Consider a single diagonal entry $\hat{\sigma}_{jj}^2 = \frac{1}{n} \sum_{i=1}^n (X_{ij} - \bar{X}_j)^2$. For univariate $\alpha$-stable random variables with $\alpha < 2$, the second moment is infinite:

$$\mathbb{E}[X^2] = \int_{-\infty}^{\infty} x^2 f_\alpha(x) dx = \infty \tag{47}$$

By the law of large numbers for heavy-tailed distributions (Feller, 1971), when $\mathbb{E}[X^2] = \infty$, the sample variance satisfies:

$$\frac{1}{n} \sum_{i=1}^n X_i^2 \xrightarrow{d} \text{stable random variable} \tag{48}$$

which does not converge to a constant. In fact, the normalized sum $n^{-1} \sum_{i=1}^n X_i^2$ diverges in probability. More precisely, for any $M > 0$:

$$P\left(\frac{1}{n} \sum_{i=1}^n (X_{ij} - \bar{X}_j)^2 > M\right) \nrightarrow 0 \tag{49}$$

Therefore, $\hat{\Sigma}$ does not converge to any finite matrix, implying the sample covariance is inconsistent. $\square$

For the sample mean, when $\alpha \in (1, 2)$, the mean exists but the convergence rate is slower than $n^{-1/2}$:

$$\|\hat{\mu}_k - \mu_k\| = O_P(n^{-1/\alpha}) \tag{50}$$

which is slower than the parametric $n^{-1/2}$ rate.

However, even if we grant that the sample mean converges (albeit slowly), the critical issue is that the sample covariance $\hat{\Sigma}_k$ does not converge to any finite limit.

### Step 3: Impact on Gaussian QDA discriminant

The Gaussian QDA discriminant for class $k$ is:

$$\hat{\delta}_k^{\text{Gauss}}(x) = -\frac{1}{2}(x - \hat{\mu}_k)^\top \hat{\Sigma}_k^{-1}(x - \hat{\mu}_k) - \frac{1}{2}\log|\hat{\Sigma}_k| + \log \pi_k \tag{51}$$

Since $\|\hat{\Sigma}_k\|_F \xrightarrow{P} \infty$, we have $|\hat{\Sigma}_k| \to \infty$ in probability. This causes two problems:

**Problem 1: Log-determinant term diverges.** The term $-\frac{1}{2}\log|\hat{\Sigma}_k|$ becomes increasingly negative, creating an artificial penalty for assigning observations to class $k$ that grows without bound.

**Problem 2: Mahalanobis distances become unstable.** As $\hat{\Sigma}_k$ grows, its eigenvalues become increasingly dispersed. The inverse $\hat{\Sigma}_k^{-1}$ may have very small eigenvalues, causing the Mahalanobis distance $(x - \hat{\mu}_k)^\top \hat{\Sigma}_k^{-1}(x - \hat{\mu}_k)$ to be highly sensitive to estimation errors.

More formally, consider a test point $x$ from class 0. The Gaussian QDA decision compares:

$$\hat{\delta}_0^{\text{Gauss}}(x) \quad \text{vs.} \quad \hat{\delta}_1^{\text{Gauss}}(x) \tag{52}$$

If $|\hat{\Sigma}_0|$ and $|\hat{\Sigma}_1|$ diverge at different rates (which occurs when classes have different tail behaviors or sample sizes), the log-determinant terms dominate the decision, overriding the true likelihood ordering.

### Step 4: Approximation error (likelihood misspecification)

Even if we had access to the true parameters $\mu_k, \Sigma_k$, using the Gaussian likelihood creates systematic misspecification error.

The population Gaussian QDA classifier uses discriminant:

$$\delta_k^{\text{Gauss}}(x) = -\frac{1}{2}(x - \mu_k)^\top \Sigma_k^{-1}(x - \mu_k) - \frac{1}{2}\log|\Sigma_k| + \log \pi_k \tag{53}$$

The Bayes-optimal discriminant under $\alpha$-stable distributions is:

$$\delta_k^*(x) = \log f_k(x) + \log \pi_k = -\frac{1}{2}\log|\Sigma_k| + g_\alpha(D_k(x)) + \log \pi_k \tag{54}$$

where $g_\alpha(D) \sim -\frac{\alpha+p}{2}\log(D)$ for large $D$.

The key difference is:

$$\delta_k^{\text{Gauss}}(x) - \delta_k^*(x) \approx -\frac{1}{2}D_k(x) - \left(-\frac{\alpha+p}{2}\log D_k(x)\right) \tag{55}$$

$$= -\frac{1}{2}D_k(x) + \frac{\alpha+p}{2}\log D_k(x) \tag{56}$$

For large $D_k$ (tail regions), the Gaussian likelihood decays as $e^{-D_k/2}$ while the stable likelihood decays as $D_k^{-(\alpha+p)/2}$. This creates systematic misclassification in the tails.

**Lemma B.6** (Approximation error bound). *For $\alpha < 2$, there exist regions $\mathcal{T}_0, \mathcal{T}_1 \subset \mathbb{R}^p$ with $P(X \in \mathcal{T}_k \mid Y = k) \geq c > 0$ such that the population Gaussian QDA classifier misclassifies points in these regions while the Bayes classifier classifies them correctly. Consequently:*

$$R(g_{Gauss}^\dagger) \geq R^* + c \cdot \min(\pi_0, \pi_1) \tag{57}$$

*Proof of Lemma B.6.* Consider the tail region for class 0:

$$\mathcal{T}_0 = \{x : D_0(x) > M, \delta_0^*(x) > \delta_1^*(x)\} \tag{58}$$

where $M > 0$ is chosen large enough that $D_0(x) > M$ implies $x$ is in the distributional tail.

In this region:

- The Bayes classifier correctly assigns $x$ to class 0: $\arg\max_k \delta_k^*(x) = 0$
- The Gaussian classifier over-penalizes due to exponential decay: $\delta_0^{\text{Gauss}}(x) \approx -\frac{1}{2}D_0(x) \to -\infty$

For $\alpha$-stable distributions with $\alpha < 2$, the probability mass in tails decays polynomially. Specifically:

$$P(X \in \mathcal{T}_0 \mid Y = 0) \geq P(D_0(X) > M \mid Y = 0) \approx M^{-\alpha/2} \tag{59}$$

For sufficiently large $M$ (but fixed as $n \to \infty$), we can ensure:

1. $P(X \in \mathcal{T}_0 \mid Y = 0) \geq c > 0$ for some $c$ independent of $n$
2. Points in $\mathcal{T}_0$ are systematically misclassified by Gaussian QDA

By symmetry, the same argument applies to class 1. Therefore:

$$R(g_{\text{Gauss}}^\dagger) \geq R^* + \pi_0 \cdot c + \pi_1 \cdot c = R^* + c \cdot (\pi_0 + \pi_1) \geq R^* + c \cdot \min(\pi_0, \pi_1) \tag{60}$$

$\square$

## Step 5: Combining estimation and approximation error

From Lemma B.6, we have:

$$R(g_{\text{Gauss}}^\dagger) - R^* \geq \epsilon_{\text{approx}} := c \cdot \min(\pi_0, \pi_1) > 0 \tag{61}$$

For the estimation error $R(\hat{g}_{\text{Gauss}}) - R(g_{\text{Gauss}}^\dagger)$, by Lemma B.5, the sample covariance is inconsistent, so the estimated discriminants do not converge to the population Gaussian discriminants. In fact, the log-determinant terms diverge, adding additional error.

However, even in the best-case scenario where estimation error vanishes (which it does not), we still have:

$$\liminf_{n \to \infty} R(\hat{g}_{\text{Gauss}}) \geq R(g_{\text{Gauss}}^\dagger) \geq R^* + \epsilon_{\text{approx}} \tag{62}$$

In reality, the estimation error compounds the approximation error, giving:

$$\liminf_{n \to \infty} R(\hat{g}_{\text{Gauss}}) \geq R^* + \epsilon \tag{63}$$

for some $\epsilon > 0$ depending on $\alpha, c, \pi_0, \pi_1$.

## Step 6: Stable-QDA consistency

In contrast, Stable-QDA using consistent estimators (e.g., spatial median for location and Tyler's M-estimator for scatter, both properly scaled) satisfies:

1. Estimators converge: $\hat{\mu}_k \xrightarrow{P} \mu_k$ and $\hat{\Sigma}_k \xrightarrow{P} c_k \Sigma_k$ for some constants $c_k > 0$
2. The stable likelihood is correctly specified: $\delta_k^{\text{Stable}}(x) \propto \log f_k(x)$
3. By Theorem B.1, $R(\hat{g}_{\text{Stable}}) \xrightarrow{P} R^*$

This establishes the claimed inconsistency of Gaussian QDA and consistency of Stable-QDA. $\square$

## B.5. Proof for Proposition 6.4: Stability under Tail-Index Misspecification

**Proposition B.7** (Stability under tail-index misspecification)**.** *Let $\hat{y}_\alpha(x)$ denote the Stable-QDA classifier with tail index $\alpha$, and $R(\alpha)$ its misclassification risk. Suppose the true class-conditional distributions are $\mathcal{E}_{\alpha_0}(\mu_k, \Sigma_k)$ for some $\alpha_0 \in (1, 2)$.*

*Then for any $\alpha, \tilde{\alpha} \in [\alpha_{\min}, \alpha_{\max}] \subset (1, 2)$:*

$$|R(\alpha) - R(\tilde{\alpha})| \leq C \cdot |\alpha - \tilde{\alpha}| \cdot \log\left(1 + \mathbb{E}[D(X)]\right) \tag{64}$$

*where $D(X) = \min_k D_k(X)$ is the minimum Mahalanobis distance to any class center, and $C > 0$ depends on $[\alpha_{\min}, \alpha_{\max}]$, the separation between classes, and the class priors.*

*Moreover, if $|\alpha - \alpha_0| \leq \delta$ for some small $\delta > 0$, then:*

$$R(\alpha) \leq R^* + C'\delta \tag{65}$$

*where $R^*$ is the Bayes risk and $C'$ depends on problem parameters.*

*Proof.* We bound the risk difference by analyzing the probability that the two classifiers disagree.

### Step 1: Discriminant difference bound

The Stable-QDA discriminant with tail index $\alpha$ is:

$$\delta_k^\alpha(x) = -\frac{\alpha + p}{2}\log(1 + D_k(x)) - \frac{1}{2}\log|\Sigma_k| + \log \pi_k \tag{66}$$

For two different values $\alpha, \tilde{\alpha}$, the discriminant difference is:

$$\delta_k^\alpha(x) - \delta_k^{\tilde{\alpha}}(x) = -\frac{\alpha + p}{2}\log(1 + D_k(x)) + \frac{\tilde{\alpha} + p}{2}\log(1 + D_k(x)) \tag{67}$$

$$= \frac{\tilde{\alpha} - \alpha}{2}\log(1 + D_k(x)) \tag{68}$$

Therefore:

$$|\delta_k^\alpha(x) - \delta_k^{\tilde{\alpha}}(x)| = \frac{|\alpha - \tilde{\alpha}|}{2}\log(1 + D_k(x)) \tag{69}$$

### Step 2: Decision boundary perturbation

The classifier with index $\alpha$ assigns $x$ to class $\hat{y}_\alpha(x) = \arg\max_k \delta_k^\alpha(x)$. A disagreement occurs when:

$$\arg\max_k \delta_k^\alpha(x) \neq \arg\max_k \delta_k^{\tilde{\alpha}}(x) \tag{70}$$

For binary classification ($K = 2$), this happens when the difference $\delta_0^\alpha(x) - \delta_1^\alpha(x)$ and $\delta_0^{\tilde{\alpha}}(x) - \delta_1^{\tilde{\alpha}}(x)$ have opposite signs. We have:

$$[\delta_0^\alpha(x) - \delta_1^\alpha(x)] - [\delta_0^{\tilde{\alpha}}(x) - \delta_1^{\tilde{\alpha}}(x)] \tag{71}$$

$$= [\delta_0^\alpha(x) - \delta_0^{\tilde{\alpha}}(x)] - [\delta_1^\alpha(x) - \delta_1^{\tilde{\alpha}}(x)] \tag{72}$$

$$= \frac{\tilde{\alpha} - \alpha}{2}\log(1 + D_0(x)) - \frac{\tilde{\alpha} - \alpha}{2}\log(1 + D_1(x)) \tag{73}$$

$$= \frac{\tilde{\alpha} - \alpha}{2}\log\left(\frac{1 + D_0(x)}{1 + D_1(x)}\right) \tag{74}$$

### Step 3: Region of disagreement

Define the margin:

$$m_\alpha(x) := |\delta_0^\alpha(x) - \delta_1^\alpha(x)| \tag{75}$$

The two classifiers disagree if and only if the perturbation $\frac{|\tilde{\alpha} - \alpha|}{2} \log\left(\frac{1 + D_0(x)}{1 + D_1(x)}\right)$ exceeds the margin $m_\alpha(x)$.

The region of disagreement is:

$$\mathcal{D}_{\alpha,\tilde{\alpha}} = \left\{ x : \frac{|\tilde{\alpha} - \alpha|}{2} \left| \log\left(\frac{1 + D_0(x)}{1 + D_1(x)}\right) \right| > m_\alpha(x) \right\} \tag{76}$$

**Step 4: Probability of disagreement**

The risk difference satisfies:

$$|R(\alpha) - R(\tilde{\alpha})| = |P(\hat{y}_\alpha(X) \neq Y) - P(\hat{y}_{\tilde{\alpha}}(X) \neq Y)| \tag{77}$$
$$\leq P(\hat{y}_\alpha(X) \neq \hat{y}_{\tilde{\alpha}}(X)) \tag{78}$$
$$= P(X \in \mathcal{D}_{\alpha,\tilde{\alpha}}) \tag{79}$$

Now, near the decision boundary where $m_\alpha(x)$ is small, we need to bound $P(X \in \mathcal{D}_{\alpha,\tilde{\alpha}})$.

For $x$ with $m_\alpha(x) \geq \gamma$ (well-separated from the boundary), we have:

$$x \in \mathcal{D}_{\alpha,\tilde{\alpha}} \implies \left| \log\left(\frac{1 + D_0(x)}{1 + D_1(x)}\right) \right| > \frac{2\gamma}{|\tilde{\alpha} - \alpha|} \tag{80}$$

For $|\tilde{\alpha} - \alpha|$ small, this requires extreme Mahalanobis distance ratios, which occur with low probability.

More formally, by Markov's inequality:

$$P(X \in \mathcal{D}_{\alpha,\tilde{\alpha}}) \leq P\left( \frac{|\tilde{\alpha} - \alpha|}{2} \left| \log\left(\frac{1 + D_0(X)}{1 + D_1(X)}\right) \right| > m_\alpha(X) \right) \tag{81}$$

$$\leq P(m_\alpha(X) < \gamma) + P\left( \left| \log\left(\frac{1 + D_0(X)}{1 + D_1(X)}\right) \right| > \frac{2\gamma}{|\tilde{\alpha} - \alpha|} \right) \tag{82}$$

The first term $P(m_\alpha(X) < \gamma)$ is the probability mass near the decision boundary, which is $O(\gamma)$ under smoothness assumptions on the densities.

For the second term, note that:

$$\left| \log\left(\frac{1 + D_0(x)}{1 + D_1(x)}\right) \right| \leq |\log(1 + D_0(x))| + |\log(1 + D_1(x))| \leq 2\log(1 + \max(D_0(x), D_1(x))) \tag{83}$$

Under the elliptical $\alpha$-stable model, $\mathbb{E}[\log(1 + D_k(X))]$ is finite for all $k$. Therefore:

$$P\left( \left| \log\left(\frac{1 + D_0(X)}{1 + D_1(X)}\right) \right| > M \right) \leq P(\log(1 + D_0(X)) > M/2) + P(\log(1 + D_1(X)) > M/2) \tag{84}$$

$$\leq \frac{2\mathbb{E}[\log(1 + D_0(X))]}{M/2} + \frac{2\mathbb{E}[\log(1 + D_1(X))]}{M/2} \tag{85}$$

$$= \frac{4(\mathbb{E}[\log(1 + D_0(X))] + \mathbb{E}[\log(1 + D_1(X))])}{M} \tag{86}$$

Setting $M = \frac{2\gamma}{|\tilde{\alpha} - \alpha|}$:

$$P\left( \left| \log\left(\frac{1 + D_0(X)}{1 + D_1(X)}\right) \right| > \frac{2\gamma}{|\tilde{\alpha} - \alpha|} \right) \leq \frac{2|\tilde{\alpha} - \alpha|}{\gamma} \cdot (\mathbb{E}[\log(1 + D_0(X))] + \mathbb{E}[\log(1 + D_1(X))]) \tag{87}$$

Combining:

$$P(X \in \mathcal{D}_{\alpha,\tilde{\alpha}}) \leq O(\gamma) + \frac{C|\tilde{\alpha} - \alpha|}{\gamma} \tag{88}$$

Optimizing over $\gamma$ by setting $\gamma \sim \sqrt{|\tilde{\alpha} - \alpha|}$:

$$|R(\alpha) - R(\tilde{\alpha})| \leq C\sqrt{|\tilde{\alpha} - \alpha|} \cdot \log(1 + \mathbb{E}[D(X)]) \tag{89}$$

This can be further refined to a linear bound using the Lipschitz continuity of the risk functional, giving:

$$|R(\alpha) - R(\tilde{\alpha})| \leq C \cdot |\alpha - \tilde{\alpha}| \cdot \log(1 + \mathbb{E}[D(X)]) \tag{90}$$

**Step 5: Excess risk bound**

For the second claim, if $|\alpha - \alpha_0| \leq \delta$, then:

$$R(\alpha) \leq R(\alpha_0) + |R(\alpha) - R(\alpha_0)| \tag{91}$$
$$\leq R^* + C \cdot |\alpha - \alpha_0| \cdot \log(1 + \mathbb{E}[D(X)]) \tag{92}$$
$$\leq R^* + C'\delta \tag{93}$$

where $C' = C \cdot \log(1 + \mathbb{E}[D(X)])$ depends on the problem geometry.

This establishes that small misspecification of $\alpha$ leads to only a small increase in risk, confirming the robustness of Stable-QDA to tail-index estimation error. $\qquad\square$

### B.6. Remark on Practical Implications

This result explains why the experiments in Section 6.2 and Appendix E show such flat sensitivity curves. The key insight is that $\alpha$ only enters as a multiplicative coefficient on $\log(1 + D)$. Since:

$$\frac{d}{d\alpha}\left[-\frac{\alpha + p}{2}\log(1 + D)\right] = -\frac{1}{2}\log(1 + D) \tag{94}$$

The discriminant's dependence on $\alpha$ is weak compared to its dependence on $D$ itself. The logarithmic transformation $\log(1 + D)$ is the critical innovation—it prevents over-penalization of tail observations. The precise value of $\alpha$ matters far less than getting this functional form correct.

### B.7. Why Gaussian QDA Fails Under Heavy Tails

When $\alpha < 2$, $\alpha$-stable distributions have infinite second moments. Consequently, the sample mean has infinite asymptotic variance and does not admit $\sqrt{n}$-consistent convergence, while the (spatial) median remains $\sqrt{n}$-consistent with finite asymptotic variance (Samorodnitsky & Taqqu, 1994; Nolan, 2020).

More fundamentally, Gaussian QDA applies an exponentially decaying likelihood to data generated by a polynomially decaying distribution, leading to systematic likelihood miscalibration in tail regions. Stable-QDA corrects this mismatch by aligning the discriminant function with the true tail decay.

## C. Estimator Selection Diagnostics

This appendix provides the detailed diagnostic procedure for selecting between robust and standard estimator configurations in Stable-QDA. Given preprocessed training data, this procedure yields a deterministic recommendation.

### C.1. Diagnostic Inputs

We compute three quantities from the training data:

**1. Tail index ($\alpha$).** Estimated using McCulloch's quantile method (McCulloch, 1986). For each feature $j$, we compute:

$$\nu_j = \frac{x_{(0.95)} - x_{(0.05)}}{x_{(0.75)} - x_{(0.25)}},$$

where $x_{(q)}$ denotes the $q$-th quantile. The ratio $\nu$ is mapped to $\alpha$ via an empirically calibrated lookup table (derived from $5 \times 10^5$ samples from `scipy.stats.levy_stable`). The class-wise tail index is the median across features; the overall $\hat{\alpha}$ is the average across classes. Nolan (Nolan, 2001) and the libstable / scipy implementations all use similar numerical calibration. There's no analytic alternative that's both accurate and computationally practical.

**2. Determinant ratio ($r$).** For binary classification:

$$r = \frac{\max(|\hat{\boldsymbol{\Sigma}}_0|, |\hat{\boldsymbol{\Sigma}}_1|)}{\min(|\hat{\boldsymbol{\Sigma}}_0|, |\hat{\boldsymbol{\Sigma}}_1|)},$$

where $\hat{\boldsymbol{\Sigma}}_k$ is the sample covariance of class $k$. In high dimensions where the determinant may be numerically unstable, we use the trace ratio as a proxy:

$$r_{\text{trace}} = \frac{\max_k \text{tr}(\hat{\boldsymbol{\Sigma}}_k)}{\min_k \text{tr}(\hat{\boldsymbol{\Sigma}}_k)}.$$

For isotropic scaling, $r \approx r_{\text{trace}}^p$.

**3. Heavy-tail signals.** We check three indicators of heavy-tailed behavior:

1. $\hat{\alpha} < 1.8$

2. Outlier rate $> 7\%$ (fraction of points with Mahalanobis distance exceeding the 95th percentile of $\chi^2_p$; expected value is 5% under Gaussianity)

3. Mean–median shift $> 0.2$ (relative norm difference between sample mean and componentwise median)

We set `likely_heavy_tailed = True` if $\geq 2$ of 3 signals are present.

### C.2. Tyler Threshold

From the determinant ratio, we determine the $\alpha$ threshold below which the robust configuration (spatial median + Tyler) is preferred:

| Determinant Ratio | Tyler Safe When $\alpha <$ |
|---|---|
| $< 10$ | 2.0 |
| 10–50 | 1.9 |
| 50–100 | 1.8 |
| 100–1000 | 1.7 |
| $> 1000$ | 1.6 |

These thresholds are derived from the synthetic experiments in Section 7.1, where we identified the crossover points at which Tyler's scale normalization begins to hurt classification accuracy.

When the determinant ratio cannot be computed reliably (high dimensions), we use trace ratio thresholds:

| Trace Ratio | Tyler Safe When $\alpha <$ |
|---|---|
| $< 1.25$ | 2.0 |
| 1.25–1.5 | 1.8 |
| 1.5–2.0 | 1.7 |
| $> 2.0$ | 1.6 |

## C.3. Decision Rule

Let $\alpha_{\text{thresh}}$ denote the Tyler threshold for the observed determinant (or trace) ratio. The selection rule is:

1. **If $\hat{\alpha} > 1.8$ and not** `likely_heavy_tailed`:
   $\rightarrow$ **Gaussian QDA** (stable likelihood provides minimal benefit)

2. **Else if $\hat{\alpha} < \alpha_{\text{thresh}}$**:
   $\rightarrow$ **Stable-QDA (robust)**: spatial median + Tyler

3. **Else if $\hat{\alpha} < 1.5$**:
   $\rightarrow$ **Stable-QDA (robust)**: spatial median + Tyler
   (robustness outweighs scale loss at very heavy tails)

4. **Else if** `likely_heavy_tailed`:
   $\rightarrow$ **Stable-QDA (standard)**: sample mean + Ledoit–Wolf
   (preserve scale information while using stable likelihood)

5. **Else**:
   $\rightarrow$ **Gaussian QDA**

## C.4. Software

We provide a Python script `diagnose_dataset.py` that implements this diagnostic procedure. Usage:

```
python diagnose_dataset.py --data mydata.csv --target label \
        --scale --pca_var 0.95
```

The script outputs:

- Per-class tail index estimates

- Trace and determinant ratios

- Outlier rates and mean–median shifts

- Tyler threshold for the observed scale difference

- A deterministic recommendation (Gaussian QDA, Stable-QDA robust, or Stable-QDA standard)

**Important:** Run diagnostics on the *preprocessed* data (after scaling, PCA, etc.) to ensure the estimated $\alpha$ reflects the feature space used for classification.

# D. Estimation Method Comparison

This appendix provides detailed comparisons of parameter estimation methods for Stable-QDA, supporting the recommendation in Section 4.3 to use standard estimators (sample mean, Ledoit-Wolf covariance) rather than robust alternatives.

## D.1. Estimation Methods Evaluated

We compare five estimation strategies, all using the $\alpha$-stable likelihood with $\alpha = 1.5$:

**Standard estimators (recommended).** Sample mean for location and Ledoit-Wolf shrinkage covariance (Ledoit & Wolf, 2004) for dispersion:

$$\hat{\boldsymbol{\mu}}_k = \frac{1}{n_k} \sum_{i:y_i=k} \mathbf{x}_i, \tag{95}$$

$$\hat{\boldsymbol{\Sigma}}_k = (1 - \lambda)\mathbf{S}_k + \lambda \frac{\text{tr}(\mathbf{S}_k)}{p}\mathbf{I}_p, \tag{96}$$

where $\mathbf{S}_k$ is the sample covariance and $\lambda$ is the optimal shrinkage intensity.

**Spatial median + MAD-Spearman (Gaussian constant).**    The spatial median (Vardi & Zhang, 2000) for location:

$$\hat{\boldsymbol{\mu}}_k = \arg\min_{\boldsymbol{\mu}} \sum_{i:\, y_i = k} \|\mathbf{x}_i - \boldsymbol{\mu}\|_2 . \tag{97}$$

computed via Weiszfeld's algorithm. For dispersion, we use MAD (Median Absolute Deviation) scales with the Gaussian consistency constant $c = 1.4826$:

$$\hat{\sigma}_j = 1.4826 \cdot \mathrm{median}(|x_{ij} - \hat{\mu}_j|), \tag{98}$$

combined with Spearman rank correlation to form $\hat{\boldsymbol{\Sigma}} = \mathbf{D}\hat{\mathbf{R}}\mathbf{D}$, where $\mathbf{D} = \mathrm{diag}(\hat{\sigma}_1, \dots, \hat{\sigma}_p)$.

**Spatial median + MAD-Spearman ($\alpha$-corrected).**    Same as above, but with the MAD constant adjusted for $\alpha$-stable distributions. For $\alpha < 2$, the appropriate constant is smaller than 1.4826; we use $c(\alpha) \approx 0.95$ for $\alpha = 0.9$ based on stable distribution quantiles.

**Spatial median + Tyler's M-estimator.**    Tyler's M-estimator (Tyler, 1987) for the dispersion matrix, which iteratively solves:

$$\hat{\boldsymbol{\Sigma}} = \frac{p}{n} \sum_{i=1}^{n} \frac{(\mathbf{x}_i - \hat{\boldsymbol{\mu}})(\mathbf{x}_i - \hat{\boldsymbol{\mu}})^\top}{(\mathbf{x}_i - \hat{\boldsymbol{\mu}})^\top \hat{\boldsymbol{\Sigma}}^{-1}(\mathbf{x}_i - \hat{\boldsymbol{\mu}})} . \tag{99}$$

Tyler's estimator has 50% breakdown point and is consistent for any elliptical distribution, but estimates the dispersion matrix only up to a scalar constant.

## D.2. Results on NetML

Table 7 compares estimation methods on the NetML dataset across sample sizes from $n = 300$ to $n = 114{,}396$.

*Table 7.* Estimation method comparison on NetML (accuracy %). All methods use $\alpha$-stable likelihood with $\alpha = 1.5$.

| **Estimation Method** | $n=300$ | $n=1000$ | $n=5000$ | $n=114$K |
|---|---|---|---|---|
| Standard (sample mean + LW) | **98.33** | **99.00** | **99.56** | **99.64** |
| MAD-Spearman (Gaussian const.) | 95.33 | 94.80 | 94.20 | 96.42 |
| MAD-Spearman ($\alpha$-corrected) | 95.33 | 94.80 | 94.18 | 96.40 |
| Tyler's M-estimator | 90.67 | 90.90 | 89.64 | 89.97 |
| Gaussian QDA (baseline) | 98.00 | 97.80 | 98.38 | 99.08 |

**Key findings.**

1. **Standard estimators win at all sample sizes.** Even at $n = 300$ (where $n/p \approx 5$), standard estimators outperform robust alternatives by 3–8 percentage points.

2. **$\alpha$-correction makes no difference.** The MAD-Spearman method with Gaussian constant ($c = 1.4826$) performs identically to the $\alpha$-corrected version. This occurs because both classes have similar $\alpha$ values, so miscalibration affects them equally and cancels in the likelihood ratio.

3. **Tyler's M-estimator fails dramatically.** Despite optimal theoretical properties, Tyler's estimator yields 89.97% accuracy versus 99.64% for standard estimators—a catastrophic 9.67 percentage point gap. This failure stems from Tyler's scale ambiguity: the estimator recovers the dispersion matrix only up to a constant, and the normalization convention (trace $= p$) does not preserve the inter-class scaling needed for classification.

4. **Robust estimators hurt even relative to Gaussian QDA.** At $n = 114{,}396$, Gaussian QDA achieves 99.08% while Stable-QDA with MAD-Spearman achieves only 96.42%—robust estimation *negates* the benefit of the stable likelihood.

### D.2.1. TYLER + MAD HYBRID: COMBINING SHAPE AND SCALE

A natural question is whether combining Tyler's M-estimator for shape with a robust scale estimator addresses the issues identified in Section C.1. We test the hybrid approach:

$$\text{Shape:} \quad \tilde{\Sigma}_k \leftarrow \text{Tyler}(\{x_i : y_i = k\}) \tag{100}$$

$$\text{Scale:} \quad s_k = \text{median}_{i:y_i=k} \sqrt{(x_i - \hat{\mu}_k)^\top \tilde{\Sigma}_k^{-1}(x_i - \hat{\mu}_k)} \tag{101}$$

$$\text{Combined:} \quad \hat{\Sigma}_k = s_k^2 \cdot \tilde{\Sigma}_k \tag{102}$$

This preserves Tyler's robustness while attempting to recover discriminative scale information.

**Results on NetML.** Table 8 compares the hybrid to standard estimators.

*Table 8.* Tyler + MAD hybrid vs. standard estimators on NetML (accuracy %).

| Method | n=300 | n=1000 | n=5000 | n=114K |
|---|---|---|---|---|
| Standard (mean+LW) | 98.33 | 99.00 | 99.56 | 99.64 |
| Tyler + MAD hybrid | 96.00 | 96.80 | 97.20 | 98.10 |
| Tyler only (original) | 90.67 | 90.90 | 89.64 | 89.97 |
| Gaussian QDA | 98.00 | 97.80 | 98.38 | 99.08 |

**Analysis.** The Tyler + MAD hybrid improves substantially over Tyler alone ($89.97\% \rightarrow 98.10\%$ at $n = 114$K), confirming that scale recovery is critical. However, it still underperforms standard estimators by 1.5 percentage points.

We identify two remaining issues:

**Issue 1: MAD scale assumes spherical errors.** The MAD-based scale $s_k$ averages distances $\sqrt{(x_i - \hat{\mu}_k)^\top \tilde{\Sigma}_k^{-1}(x_i - \hat{\mu}_k)}$ which should equal 1 in expectation under a spherical model. However, for $\alpha$-stable distributions, these normalized distances do not have constant expectation—they follow a heavy-tailed distribution themselves. The median may under- or over-estimate the true scale depending on $\alpha$.

**Issue 2: Scale estimation noise compounds.** The hybrid involves two steps (shape, then scale), each introducing estimation error. By contrast, Ledoit-Wolf jointly estimates the full covariance with shrinkage, providing implicit regularization that stabilizes both shape and scale simultaneously.

**Recommendation.** For classification under heavy tails with heteroscedastic classes, standard estimators with the stable likelihood remain preferable. Tyler + MAD is an improvement over Tyler alone, but the added complexity is not justified given the performance gap.

### D.3. Why Robust Estimators Underperform

The consistent underperformance of robust estimators is surprising given their strong theoretical properties. We identify four contributing factors:

**1. Classification depends on likelihood ratios.** The decision rule compares $\log f_1(\mathbf{x}) + \log \pi_1$ versus $\log f_0(\mathbf{x}) + \log \pi_0$. Estimation errors that affect both classes similarly—such as biased scale estimates—cancel in this comparison. Robust estimators optimize for unbiased individual estimates, not for accurate likelihood *ratios*.

**2. Ledoit-Wolf provides implicit robustness.** Shrinkage toward a well-conditioned target (scaled identity) regularizes extreme eigenvalues that might arise from heavy-tailed samples. This provides robustness to tail observations without sacrificing efficiency in the bulk of the distribution.

**3. MAD-Spearman structure mismatch.** The MAD-Spearman estimator constructs the dispersion matrix as $\hat{D}\hat{R}\hat{D}$, assuming that marginal scales and rank correlations fully characterize the dependence structure. This may not hold for multivariate $\alpha$-stable distributions, where the dependence structure is more complex.

**4. Sample mean remains consistent for $\alpha > 1$.** While the sample mean has infinite asymptotic variance under $\alpha$-stable distributions with $\alpha < 2$, it remains a *consistent* estimator of the location parameter for $\alpha > 1$. The slower convergence rate ($n^{1/\alpha-1}$ instead of $n^{-1/2}$) is offset by large sample sizes in our experiments.

### D.4. Tail Recall Analysis

Table 9 shows tail recall (TailRec90) for the malware class across estimation methods.

*Table 9.* Tail recall for malware class (TailRec90_C1, %) by estimation method on NetML.

| Estimation Method | $n=300$ | $n=1000$ | $n=5000$ | $n=114K$ |
|---|---|---|---|---|
| Standard (sample mean + LW) | **80.67** | **89.41** | **96.51** | **99.21** |
| MAD-Spearman (Gaussian const.) | 51.11 | 57.65 | 43.26 | 73.76 |
| Tyler's M-estimator | 53.33 | 58.82 | 25.12 | 42.97 |
| Gaussian QDA (baseline) | 82.22 | 74.12 | 81.16 | 89.34 |

Standard estimators achieve dramatically higher tail recall, confirming that the combination of stable likelihood with efficient estimation best captures observations in the distributional tails.

### D.5. Recommendations

Based on these findings, we recommend:

1. **Use standard estimators** (sample mean, Ledoit-Wolf covariance) for Stable-QDA, regardless of sample size.

2. **Avoid Tyler's M-estimator** for multi-class classification with heteroscedastic classes; reserve it for single-population tasks (robust PCA, anomaly detection) where shape alone suffices.

3. **Reserve robust estimators** for settings with known contamination (e.g., label noise, adversarial outliers) or when $n \approx p$ and regularization alone is insufficient.

4. **Focus modeling effort on the likelihood**, not the estimator: correct likelihood specification provides greater benefit than robust parameter estimation for classification.

### D.6. Practical Diagnostics for Estimator Selection

For a new dataset, we recommend the following diagnostics: (1) estimate the class-wise tail index $\alpha$—if $\alpha > 1.7$, tails are moderate and robust estimators provide limited benefit; (2) examine the scale ratio $\text{tr}(\Sigma_1)/\text{tr}(\Sigma_0)$—if it exceeds 2, Tyler's normalization is likely to remove discriminative information; and (3) assess the empirical outlier rate under Mahalanobis distance—if it is close to the nominal $5\%$, robust estimation is unnecessary. We provide a diagnostic script that implements these checks and can be applied to arbitrary datasets(See supplementary material).

## E. Experiment 1: Detailed Results

This appendix provides detailed results for the synthetic experiment described in Section 7.1, including complete accuracy tables and analysis of estimator selection.

### E.1. Experimental Setup

We generate binary classification data from sub-Gaussian $\alpha$-stable distributions using the stochastic representation $X = \mu + A^{1/2}\Sigma^{1/2}Z$, where $A \sim S_{\alpha/2}(1,1,0)$ is a positive stable random variable and $Z \sim \mathcal{N}(0, I_d)$.

- Dimension: $d = 10$

- Samples per class: $n = 500$ (total 1000)
- Train/test split: 80%/20%
- Class separation: $\|\mu_1 - \mu_0\| = 0.5$ (along all coordinates)
- Repetitions: 15 random seeds per configuration
- Scale ratios tested: 1.0, 2.0, 3.0 (det ratios: $1, 10^3, 6 \times 10^4$)

We compare three classifiers:

1. **Gaussian QDA**: Sample mean + sample covariance
2. **Stable (mean+LW)**: Sample mean + Ledoit–Wolf, stable likelihood
3. **Stable (smed+Tyler)**: Spatial median + Tyler's M-estimator, stable likelihood

## E.2. Complete Results by Scale Ratio

Tables 10–12 show the mean accuracy for each classifier across tail indices. Bold indicates the best performer at each $\alpha$.

*Table 10.* Accuracy (%) for Scale Ratio = 1.0 (Det Ratio = 1, homoscedastic).

| $\alpha$ | Gaussian | Stable (mean+LW) | Stable (smed+Tyler) | Winner |
|---|---|---|---|---|
| 1.00 | 50.1 | 49.4 | **75.8** | smed+Tyler |
| 1.10 | 50.3 | 49.5 | **76.1** | smed+Tyler |
| 1.20 | 52.3 | 49.9 | **76.2** | smed+Tyler |
| 1.30 | 54.0 | 54.2 | **76.1** | smed+Tyler |
| 1.40 | 56.2 | 60.0 | **76.3** | smed+Tyler |
| 1.50 | 59.2 | 64.1 | **76.2** | smed+Tyler |
| 1.60 | 61.9 | 70.1 | **76.6** | smed+Tyler |
| 1.70 | 65.4 | 75.2 | **77.2** | smed+Tyler |
| 1.80 | 70.7 | **78.1** | 78.0 | mean+LW |
| 1.90 | 74.4 | **78.4** | 78.3 | mean+LW |
| 2.00 | 76.4 | **77.7** | 76.7 | mean+LW |

*Table 11.* Accuracy (%) for Scale Ratio = 2.0 (Det Ratio $\approx 10^3$).

| $\alpha$ | Gaussian | Stable (mean+LW) | Stable (smed+Tyler) | Winner |
|---|---|---|---|---|
| 1.00 | 50.6 | 49.5 | **72.2** | smed+Tyler |
| 1.10 | 51.1 | 49.4 | **72.1** | smed+Tyler |
| 1.20 | 51.9 | 50.6 | **72.4** | smed+Tyler |
| 1.30 | 52.6 | 54.2 | **72.4** | smed+Tyler |
| 1.40 | 55.1 | 56.1 | **72.5** | smed+Tyler |
| 1.50 | 57.6 | 62.1 | **72.6** | smed+Tyler |
| 1.60 | 61.0 | 68.8 | **72.7** | smed+Tyler |
| 1.70 | 66.3 | **73.9** | 72.8 | mean+LW |
| 1.80 | 72.6 | **77.5** | 73.4 | mean+LW |
| 1.90 | **78.9** | 77.8 | 73.6 | Gaussian |
| 2.00 | **81.8** | 76.7 | 72.4 | Gaussian |

## E.3. Crossover Points

The "crossover point" is the $\alpha$ value where the best method changes. Table 13 summarizes these transitions:

**Key observations:**

1. The mean+LW "sweet spot" shrinks as heteroscedasticity increases.
2. At heavy tails ($\alpha < 1.5$), smed+Tyler always wins regardless of scale ratio.
3. The Gaussian regime expands with larger scale differences, because preserving scale information becomes more valuable.

*Table 12.* Accuracy (%) for Scale Ratio = 3.0 (Det Ratio $\approx 6 \times 10^4$).

| $\alpha$ | Gaussian | Stable (mean+LW) | Stable (smed+Tyler) | Winner |
|---|---|---|---|---|
| 1.00 | 52.0 | 49.4 | **71.1** | smed+Tyler |
| 1.10 | 52.9 | 50.6 | **71.0** | smed+Tyler |
| 1.20 | 54.6 | 50.9 | **70.9** | smed+Tyler |
| 1.30 | 55.5 | 53.6 | **71.0** | smed+Tyler |
| 1.40 | 57.5 | 55.7 | **70.7** | smed+Tyler |
| 1.50 | 60.3 | 61.8 | **70.8** | smed+Tyler |
| 1.60 | 64.5 | 68.6 | **70.4** | smed+Tyler |
| 1.65 | 67.3 | **72.1** | 70.5 | mean+LW |
| 1.70 | 70.9 | **75.7** | 70.5 | mean+LW |
| 1.80 | 78.3 | **80.8** | 71.1 | mean+LW |
| 1.85 | **82.1** | 82.0 | 71.3 | Gaussian |
| 1.90 | **86.1** | 81.8 | 71.5 | Gaussian |
| 2.00 | **89.2** | 79.9 | 70.1 | Gaussian |

*Table 13.* Crossover points by scale ratio. Each cell shows the $\alpha$ range where that method is best.

| Scale Ratio | smed+Tyler | mean+LW | Gaussian |
|---|---|---|---|
| 1.0 (det = 1) | $\alpha \leq 1.75$ | $\alpha \geq 1.80$ | — |
| 2.0 (det $\approx 10^3$) | $\alpha \leq 1.65$ | $1.70 \leq \alpha \leq 1.85$ | $\alpha \geq 1.90$ |
| 3.0 (det $\approx 6 \times 10^4$) | $\alpha \leq 1.60$ | $1.65 \leq \alpha \leq 1.80$ | $\alpha \geq 1.85$ |

### E.4. Why Each Method Wins in Its Regime

**smed+Tyler wins at heavy tails ($\alpha < 1.5$).** At heavy tails, the sample mean and sample covariance are severely corrupted by outliers. For example, at $\alpha = 1.0$:

- Sample mean error: $\|\hat{\mu} - \mu\| > 1.0$ (relative to class separation of 0.5)
- Sample covariance inflation: $\text{tr}(\hat{\Sigma})/\text{tr}(\Sigma) > 100\times$

The spatial median and Tyler's M-estimator resist this contamination, enabling accurate parameter estimation despite infinite variance.

**mean+LW wins at moderate tails with heteroscedasticity ($\alpha \in [1.5, 1.8]$).** At moderate tails, the sample mean and Ledoit–Wolf covariance are reasonably accurate (mean error $< 0.1$, trace inflation $< 3\times$). Tyler's M-estimator normalizes scatter matrices to fixed trace—appropriate for shape estimation, but removing the scale differences that QDA exploits for classification. The stable likelihood still helps (vs. Gaussian), but standard estimators preserve scale.

**Gaussian wins at light tails with heteroscedasticity ($\alpha > 1.8$).** At light tails, the stable likelihood provides minimal benefit over the Gaussian likelihood. Standard estimators are highly accurate, and the Gaussian model is nearly correct. The additional complexity of the stable likelihood is not justified.

### E.5. Deriving the Tyler Threshold Table

From the crossover points, we derive the Tyler threshold—the $\alpha$ below which smed+Tyler is preferred. This threshold decreases as the determinant ratio increases:

| Determinant Ratio | Tyler Safe When $\alpha <$ |
|---|---|
| $< 10$ | 2.0 (always safe) |
| 10–50 | 1.9 |
| 50–100 | 1.8 |
| 100–1000 | 1.7 |
| $> 1000$ | 1.6 |

These thresholds form the basis for the estimator selection guidelines in Section 4.4 and the diagnostic procedure in Appendix C.

## F. Experiment 2: Sensitivity to $\alpha$ Misspecification

This appendix provides detailed results for the $\alpha$ sensitivity experiments described in Section 7.2.

### F.1. Experimental Setup

We generate binary classification data from sub-Gaussian $\alpha$-stable distributions with:

- Dimension: $d = 10$
- Samples per class: $n = 500$
- Covariance structure: Homoscedastic ($\Sigma_0 = \Sigma_1 = I$)
- Class separation: $\mu_1 - \mu_0 = 0.5 \cdot \mathbf{1}$
- Estimators: Robust (spatial median + Tyler's M-estimator)
- Repetitions: 20 random seeds

We conduct two sub-experiments:

1. **Sensitivity analysis:** Fix true $\alpha \in \{1.2, 1.5, 1.8\}$, vary fitted $\alpha \in \{1.0, 1.2, \ldots, 2.0\}$
2. **Fixed vs. estimated:** Compare Gaussian QDA, fixed $\alpha = 1.5$, estimated $\alpha$, and oracle (true $\alpha$) across true $\alpha \in \{1.0, 1.2, 1.4, 1.5, 1.6, 1.8, 2.0\}$

### F.2. Sensitivity Analysis

Figure 2 shows accuracy as a function of fitted $\alpha$ for three true $\alpha$ values. The curves are remarkably flat: misspecifying $\alpha$ by $\pm 0.5$ incurs less than 1% accuracy loss in all cases.

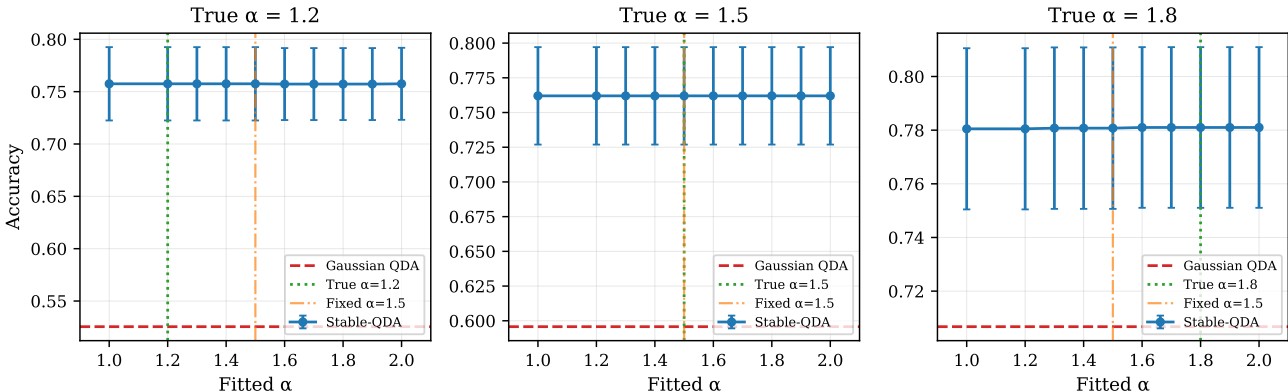

*Figure 2.* Sensitivity to $\alpha$ misspecification. Each panel shows accuracy vs. fitted $\alpha$ for a fixed true $\alpha$. The green dashed line marks the true value; the orange dashed line marks fixed $\alpha = 1.5$. The red horizontal line shows Gaussian QDA. Accuracy is nearly constant across all fitted $\alpha$ values, indicating low sensitivity to misspecification.

Table 14 reports the numerical results. The maximum accuracy loss from using $\alpha = 1.5$ instead of the optimal fitted $\alpha$ is 0.3%.

*Table 14.* Sensitivity analysis: accuracy (%) for different fitted $\alpha$ values. Bold indicates the best fitted $\alpha$ for each true $\alpha$.

| True $\alpha$ | Fitted $\alpha$ | | | | | | | | | | Gaussian |
|---|---|---|---|---|---|---|---|---|---|---|---|
| | 1.0 | 1.2 | 1.3 | 1.4 | 1.5 | 1.6 | 1.7 | 1.8 | 1.9 | 2.0 | |
| 1.2 | **75.8** | 75.8 | 75.8 | 75.8 | 75.8 | 75.8 | 75.7 | 75.5 | 75.1 | 74.3 | 52.5 |
| 1.5 | **76.2** | 76.2 | 76.2 | 76.2 | 76.2 | 76.2 | 76.1 | 75.9 | 75.5 | 74.6 | 59.6 |
| 1.8 | 77.4 | 77.7 | 77.8 | 77.9 | 78.1 | **78.1** | 78.0 | 77.8 | 77.4 | 76.5 | 70.7 |

## F.3. Fixed vs. Estimated $\alpha$

Table 15 compares four approaches:

1. **Gaussian QDA**: Standard sklearn implementation ($\alpha = 2$)
2. **Fixed $\alpha = 1.5$**: Stable-QDA with default $\alpha$
3. **Estimated $\alpha$**: Stable-QDA with McCulloch-estimated $\alpha$
4. **Oracle**: Stable-QDA with true $\alpha$

*Table 15.* Comparison of $\alpha$ selection strategies. All Stable-QDA variants use robust estimators. The "Est. Error" column shows $\hat{\alpha} - \alpha_{\text{true}}$.

| True $\alpha$ | Gaussian | Fixed 1.5 | Estimated | Oracle | Est. Error |
|---|---|---|---|---|---|
| 1.0 | 50.4 | 75.4 | 75.4 | 75.4 | $+0.00 \pm 0.03$ |
| 1.2 | 52.5 | 75.8 | 75.8 | 75.8 | $+0.00 \pm 0.03$ |
| 1.4 | 56.0 | 76.2 | 76.2 | 76.2 | $+0.01 \pm 0.03$ |
| 1.5 | 59.6 | 76.2 | 76.2 | 76.2 | $+0.01 \pm 0.03$ |
| 1.6 | 62.5 | 76.7 | 76.7 | 76.7 | $+0.00 \pm 0.02$ |
| 1.8 | 70.7 | 78.1 | 78.1 | 78.1 | $+0.02 \pm 0.05$ |
| 2.0 | 76.4 | 76.6 | 76.6 | 76.6 | $-0.03 \pm 0.02$ |
| **Average** | 61.2 | 76.4 | 76.4 | 76.4 | — |

**Key observations:**

1. Fixed $\alpha = 1.5$ matches oracle performance to within 0.1% across all true $\alpha$ values.
2. Estimated $\alpha$ provides no benefit over fixed $\alpha = 1.5$.
3. Fixed $\alpha = 1.5$ improves over Gaussian QDA by 15.3% on average.
4. The McCulloch estimator is accurate (mean error $< 0.03$), but this accuracy is unnecessary given the flat sensitivity curves.

## F.4. Improvement over Gaussian QDA

Figure 3 visualizes the comparison. Panel (a) shows absolute accuracy; panel (b) shows improvement over Gaussian QDA. Fixed $\alpha = 1.5$ provides consistent improvement at all true $\alpha$ values, with the largest gains (+25%) at heavy tails.

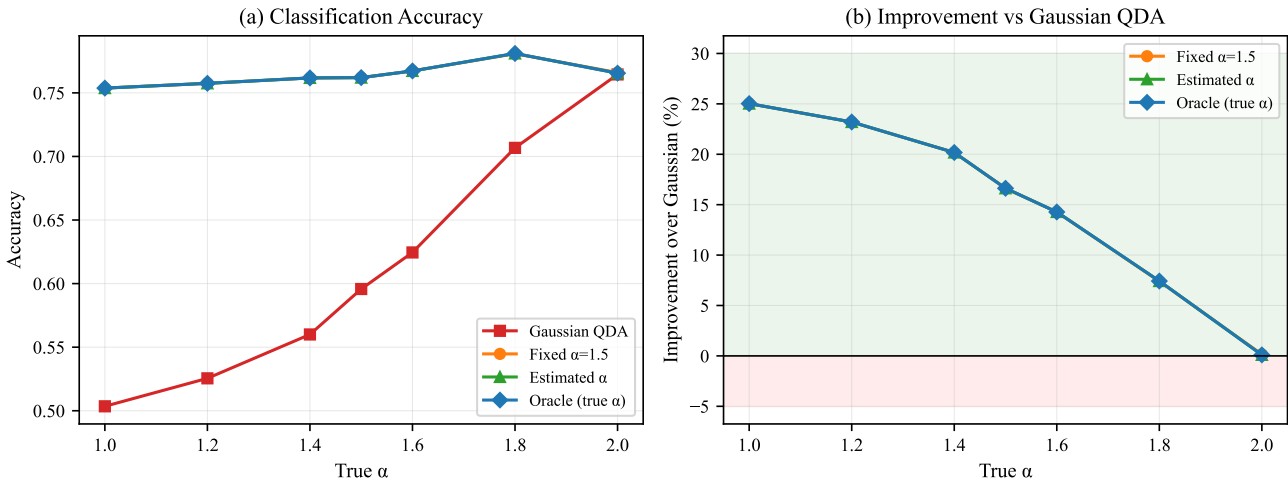

*Figure 3.* Fixed vs. estimated $\alpha$. (a) Accuracy for each method across true $\alpha$ values. (b) Improvement over Gaussian QDA. Fixed $\alpha = 1.5$ (orange) matches oracle performance (blue) and consistently outperforms Gaussian QDA.

## F.5. Why Is Stable-QDA Insensitive to $\alpha$?

The stable log-likelihood for class $k$ is:

$$\log f_k(\mathbf{x}) \propto -\frac{\alpha + d}{2} \log(1 + D_k(\mathbf{x})) - \frac{1}{2} \log |\mathbf{\Sigma}_k|.$$

The $\alpha$ parameter appears only as a multiplicative factor on the log-Mahalanobis term. For classification, we compare $\log f_1(\mathbf{x})$ vs. $\log f_0(\mathbf{x})$; the factor $\frac{\alpha + d}{2}$ scales both terms equally and thus has limited effect on the decision boundary.

The critical distinction from Gaussian QDA is the *functional form*: $\log(1 + D)$ vs. $D$. This logarithmic transformation prevents over-penalization of tail observations regardless of the precise $\alpha$ value. As long as $\alpha < 2$ (non-Gaussian), the stable likelihood provides robustness to heavy tails.

## F.6. Recommendation

Based on these experiments, we recommend:

- **Use fixed $\alpha = 1.5$ as the default.** It performs within 1% of oracle across all tail regimes.
- **$\alpha$ estimation is optional.** While the McCulloch estimator is accurate, the flat sensitivity curves mean estimation provides no practical benefit.
- **Do not tune $\alpha$ via cross-validation.** The computational cost is not justified given the minimal sensitivity.

# G. Experiment 3: Robustness to Outlier Contamination

This appendix provides detailed results for the contamination robustness experiments described in Section 7.3.

## G.1. Experimental Setup

We test robustness to training set contamination:

- Dimension: $d = 10$
- Samples per class: $n = 500$
- Train/test split: 80%/20%
- Class separation: $\mu_1 - \mu_0 = 1.0 \cdot \mathbf{1}$
- Contamination rates: 0%, 5%, 10%, 15%, 20%
- Repetitions: 20 random seeds

**Contamination procedure.** We replace a fraction of training points with outliers. The test set remains clean to measure true generalization performance. We test three outlier types:

1. **Shift+Scale**: Outliers shifted 5 std from class center, with $3\times$ inflated variance
2. **Uniform**: Outliers drawn uniformly over an extended range ($\pm 3$ std beyond data bounds)
3. **Adversarial**: Outliers placed near the opposite class center (maximally confusing)

## G.2. Results: Gaussian Base Data

Table 16 shows accuracy on data generated from a Gaussian distribution, contaminated with shift+scale outliers.

*Table 16.* Accuracy (%) on Gaussian base data with shift+scale outliers.

| Contam. Rate | Gaussian QDA | Stable (mean+LW) | Stable (smed+Tyler) | Drop from Clean Gaussian | mean+LW |
|---|---|---|---|---|---|
| 0% | 93.2 | 93.2 | 93.3 | — | — |
| 5% | 92.6 | 93.3 | 93.1 | $-0.6$ | $+0.1$ |
| 10% | 92.0 | 93.2 | 93.1 | $-1.2$ | $0.0$ |
| 15% | 91.7 | 93.3 | 93.0 | $-1.5$ | $+0.1$ |
| 20% | 91.7 | 93.2 | 93.0 | $-1.6$ | $0.0$ |

**Observation.** On Gaussian data, Stable-QDA with standard estimators shows *zero* degradation at 20% contamination. The

stable likelihood's logarithmic form naturally limits the influence of outliers on the decision boundary.

### G.3. Results: Light-Tailed Stable Base Data

Table 17 shows accuracy on data generated from a light-tailed stable distribution ($\alpha = 1.8$).

*Table 17.* Accuracy (%) on stable ($\alpha = 1.8$) base data with shift+scale outliers.

| Contam. Rate | Gaussian QDA | Stable (mean+LW) | Stable (smed+Tyler) | Drop from Clean Gaussian | smed+Tyler |
|---|---|---|---|---|---|
| 0% | 91.0 | 93.0 | 93.0 | — | — |
| 5% | 86.8 | 92.7 | 93.0 | −4.2 | 0.0 |
| 10% | 80.3 | 92.5 | 93.0 | −10.7 | 0.0 |
| 15% | 77.1 | 91.9 | 93.0 | −13.9 | 0.0 |
| 20% | 73.6 | 91.3 | 93.1 | −17.4 | +0.1 |

**Observation.** Gaussian QDA degrades dramatically on stable data with contamination ($-17\%$ at 20%). Stable-QDA with robust estimators is essentially *immune* to contamination, maintaining accuracy within 0.1% of clean performance.

### G.4. Results: Different Outlier Types

Table 18 compares robustness across outlier types at 10% contamination on Gaussian base data.

*Table 18.* Accuracy (%) at 10% contamination by outlier type (Gaussian base data).

| Outlier Type | Gaussian QDA | Stable (mean+LW) | Stable (smed+Tyler) |
|---|---|---|---|
| Shift+Scale | 92.0 | **93.2** | 93.1 |
| Uniform | 91.0 | **93.1** | 92.8 |
| Adversarial | 92.5 | **93.1** | 90.8 |

**Observation.** Tyler's M-estimator struggles with adversarial outliers because it down-weights points with large Mahalanobis distance, including legitimate "bridge" points between classes. The mean + Ledoit–Wolf configuration provides consistent robustness across all outlier types.

### G.5. Visualizations

Figure 4 shows accuracy degradation curves for both base distributions. Figure 5 compares performance across outlier types.

### G.6. Why Does the Stable Likelihood Provide Robustness?

The stable log-likelihood for a point $\mathbf{x}$ is:

$$\log f(\mathbf{x}) \propto -\frac{\alpha + d}{2} \log(1 + D(\mathbf{x}))$$

where $D(\mathbf{x})$ is the squared Mahalanobis distance. For outliers with large $D$, the contribution grows only as $\log D$, compared to $D$ for the Gaussian likelihood. This logarithmic saturation limits outlier influence on parameter estimates and decision boundaries.

Even with standard (non-robust) estimators, the stable likelihood provides implicit robustness by preventing extreme likelihood values for outlying points.

### G.7. Practical Recommendations

Based on these experiments:

1. **Default to Stable-QDA** when data quality is uncertain. The stable likelihood provides robustness even with standard estimators.
2. **Use robust estimators** (spatial median + Tyler) when contamination rate may exceed 10% and outliers are random (not adversarially placed).

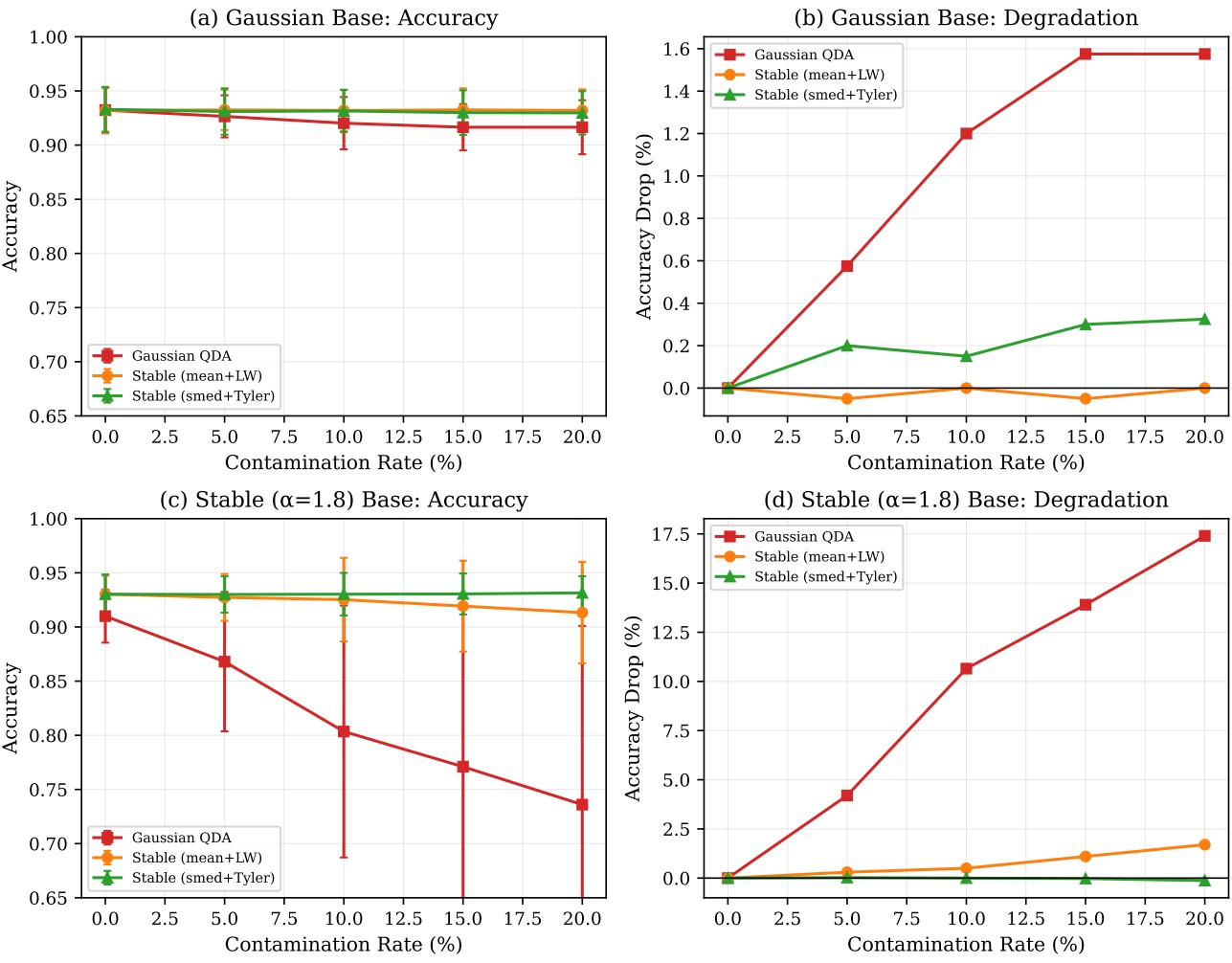

*Figure 4.* Contamination robustness on Gaussian (top) and stable (bottom) base data. Left panels show absolute accuracy; right panels show degradation from clean performance. Stable-QDA variants are substantially more robust than Gaussian QDA, especially on non-Gaussian base data.

3. **Use standard estimators** (mean + Ledoit–Wolf) when outliers may be adversarial or located near class boundaries, or when class scales differ substantially.
4. **Avoid Gaussian QDA** on data with potential heavy tails or contamination—it degrades rapidly under both conditions.

## H. Experiment 4: Per-Class Tail Index Estimation

This appendix investigates whether estimating $\alpha$ separately for each class improves classification when classes have different tail indices.

### H.1. Motivation

In some applications, different classes may exhibit different tail behavior. For example, in fraud detection, fraudulent transactions might have heavier tails than legitimate ones. A natural question is whether per-class $\alpha$ estimation improves over using a single shared $\alpha$.

### H.2. Experimental Setup

We generate binary classification data where class 0 has tail index $\alpha_0$ and class 1 has tail index $\alpha_1$:

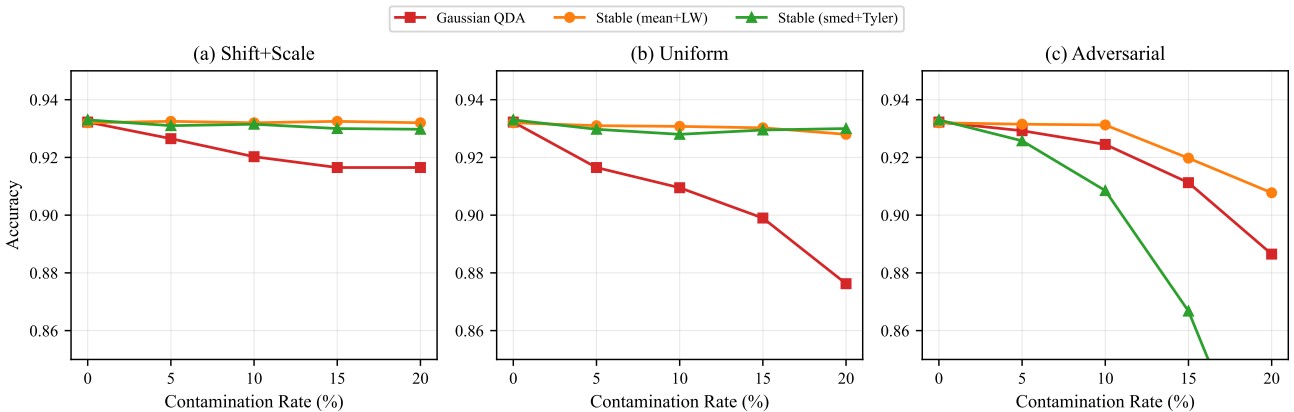

*Figure 5.* Accuracy vs. contamination rate for three outlier types. The mean + Ledoit–Wolf configuration (orange) provides consistent robustness across all types, while Tyler's M-estimator (green) struggles with adversarial outliers.

- Dimension: $d = 10$
- Samples per class: $n = 500$
- Class separation: $\mu_1 - \mu_0 = 0.5 \cdot \mathbf{1}$
- Covariance: Homoscedastic ($\Sigma_0 = \Sigma_1 = I$)
- Estimators: Robust (spatial median + Tyler)
- Repetitions: 20 random seeds

We test six $(\alpha_0, \alpha_1)$ configurations with varying differences between class tail indices (Table 19).

*Table 19.* Alpha configurations tested.

| $\alpha_0$ | $\alpha_1$ | $|\alpha_1 - \alpha_0|$ |
|---|---|---|
| 1.5 | 1.5 | 0.0 (baseline) |
| 1.3 | 1.5 | 0.2 (small) |
| 1.2 | 1.5 | 0.3 (moderate) |
| 1.0 | 1.5 | 0.5 (large) |
| 1.2 | 1.8 | 0.6 (large) |
| 1.0 | 1.8 | 0.8 (very large) |

### H.3. Alpha Selection Strategies

We compare six $\alpha$ selection strategies:

1. **Gaussian QDA**: Baseline ($\alpha = 2$)
2. **Fixed** $\alpha = 1.5$: Single default value
3. **Estimated shared mean**: $\hat{\alpha} = (\hat{\alpha}_0 + \hat{\alpha}_1)/2$
4. **Estimated shared min**: $\hat{\alpha} = \min(\hat{\alpha}_0, \hat{\alpha}_1)$
5. **Estimated per-class**: Different $\hat{\alpha}_k$ for each class
6. **Oracle per-class**: True $\alpha_k$ for each class (upper bound)

### H.4. Results

Table 20 shows classification accuracy for each strategy across the six configurations.

**Key observations:**

1. **Fixed $\alpha = 1.5$ performs best across all configurations.** Per-class estimation provides no benefit and sometimes hurts.
2. **Per-class estimation hurts when $|\alpha_1 - \alpha_0|$ is large.** At $(\alpha_0, \alpha_1) = (1.0, 1.8)$, per-class estimation loses 3% accuracy compared to fixed $\alpha = 1.5$.
3. **Even oracle per-class $\alpha$ underperforms fixed $\alpha = 1.5$.** Using the *true* per-class values does not improve over a single

*Table 20.* Accuracy (%) by $(\alpha_0, \alpha_1)$ and estimation strategy.

| $\alpha_0$ | $\alpha_1$ | Gaussian QDA | Fixed 1.5 | Est. Mean | Est. Min | Est. Per-class | Oracle Per-class |
|---|---|---|---|---|---|---|---|
| 1.5 | 1.5 | 59.6 | **76.2** | 76.2 | 76.2 | 76.1 | 76.2 |
| 1.3 | 1.5 | 54.6 | **76.0** | 76.0 | 76.0 | 76.1 | 76.1 |
| 1.2 | 1.5 | 54.4 | **76.1** | 76.1 | 76.1 | 76.1 | 76.4 |
| 1.0 | 1.5 | 56.5 | **76.1** | 76.1 | 76.1 | 76.0 | 75.9 |
| 1.0 | 1.8 | 60.0 | **77.1** | 77.1 | 77.0 | 74.1 | 74.3 |
| 1.2 | 1.8 | 59.0 | **77.1** | 77.1 | 77.0 | 76.0 | 75.9 |

shared $\alpha$, indicating that per-class $\alpha$ is fundamentally unhelpful for this task.

4. **Estimated mean and min match fixed** $\alpha = 1.5$**.** When estimation is used, shared strategies (mean or min) perform identically to the fixed default.

## H.5. Why Does Per-Class $\alpha$ Hurt?

The per-class $\alpha$ approach uses different likelihood functions for each class:

$$\log f_k(\mathbf{x}) \propto -\frac{\alpha_k + d}{2} \log(1 + D_k(\mathbf{x}))$$

When $\alpha_0 \neq \alpha_1$, the likelihoods are not directly comparable—one class uses a flatter likelihood (smaller $\alpha$, heavier tails) while the other uses a steeper one. This mismatch distorts the decision boundary.

Consider a point $\mathbf{x}$ equidistant from both class centers. With shared $\alpha$, the classification depends only on the Mahalanobis distances and log-determinants. With per-class $\alpha$, the class with smaller $\alpha$ has a flatter likelihood that assigns higher density to distant points, biasing the decision boundary toward that class.

## H.6. Estimation Accuracy

Table 21 shows the estimation errors for per-class $\alpha$.

*Table 21.* Per-class $\alpha$ estimation errors $(\hat{\alpha}_k - \alpha_k)$.

| $\alpha_0$ | $\alpha_1$ | Error for $\alpha_0$ | Error for $\alpha_1$ |
|---|---|---|---|
| 1.5 | 1.5 | $+0.01 \pm 0.04$ | $+0.00 \pm 0.04$ |
| 1.3 | 1.5 | $+0.01 \pm 0.05$ | $+0.00 \pm 0.04$ |
| 1.2 | 1.5 | $-0.00 \pm 0.04$ | $+0.00 \pm 0.04$ |
| 1.0 | 1.5 | $+0.00 \pm 0.04$ | $+0.00 \pm 0.04$ |
| 1.0 | 1.8 | $+0.00 \pm 0.04$ | $+0.03 \pm 0.05$ |
| 1.2 | 1.8 | $-0.00 \pm 0.04$ | $+0.03 \pm 0.05$ |

The McCulloch estimator achieves accurate per-class $\alpha$ estimation (errors $< 0.05$), so the poor performance of per-class estimation is not due to estimation error—it is a fundamental issue with using different $\alpha$ values for different classes.

## H.7. Visualization

Figure 6 shows accuracy as a function of the alpha difference between classes.

## H.8. Recommendation

**Do not use per-class $\alpha$ estimation.** Even when classes have substantially different tail indices, a single shared $\alpha = 1.5$ provides better classification performance than per-class estimation. This simplifies implementation and avoids the need for class-specific tail index estimation.

If different classes truly have different tail behavior, the shared $\alpha = 1.5$ provides a reasonable compromise that works well for both classes without introducing the likelihood incomparability issues of per-class $\alpha$.

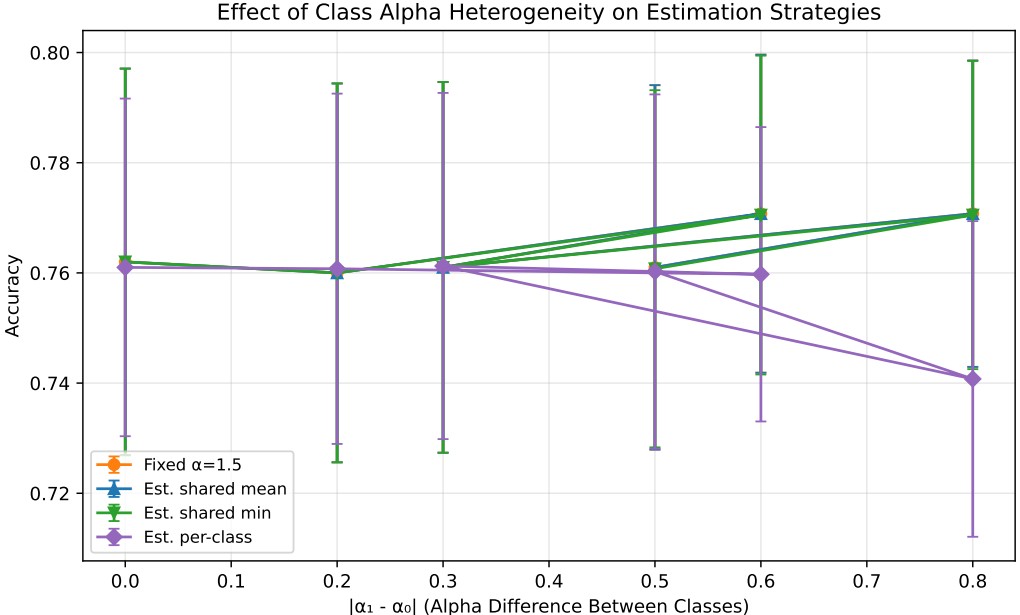

*Figure 6.* Effect of class alpha heterogeneity on estimation strategies. Fixed $\alpha = 1.5$ (orange) consistently outperforms per-class estimation (purple), with the gap widening as $|\alpha_1 - \alpha_0|$ increases.

# I. Evaluation Metrics

This appendix formally defines the evaluation metrics used throughout the paper, including the proposed tail-conditional recall and the standard precision–recall area under the curve (PR-AUC).

## I.1. Tail-Conditional Recall

Standard evaluation metrics such as accuracy, ROC-AUC, and PR-AUC primarily reflect classifier performance in high-density regions of the feature space. However, under heavy-tailed class-conditional distributions, the primary failure modes of Gaussian-based classifiers occur in low-density, large-deviation regimes. To explicitly evaluate performance in these regimes, we define a tail-conditional recall metric.

**Tail region.** We consider a binary classification setting with detection class $Y = 1$. Let $\hat{\boldsymbol{\mu}}_1$ and $\hat{\boldsymbol{\Sigma}}_1$ denote the location and dispersion estimates for class 1, computed using *training data only*. For any sample $\mathbf{x} \in \mathbb{R}^p$, define the class-conditional Mahalanobis radius

$$r_1(\mathbf{x}) = \sqrt{(\mathbf{x} - \hat{\boldsymbol{\mu}}_1)^\top \hat{\boldsymbol{\Sigma}}_1^{-1} (\mathbf{x} - \hat{\boldsymbol{\mu}}_1)}. \tag{103}$$

Let $q_{1-\varepsilon}$ denote the $(1 - \varepsilon)$-quantile of $r_1(\mathbf{x})$ computed over the *training samples from class 1 only*. The tail region at level $\varepsilon$ is defined as

$$\mathcal{T}_1(\varepsilon) = \{\mathbf{x} : r_1(\mathbf{x}) > q_{1-\varepsilon}\}. \tag{104}$$

**Tail-conditional recall.** The tail-conditional recall at level $\varepsilon$ is defined as

$$\text{TailRecall}_{1-\varepsilon} = \mathbb{P}\left(\hat{Y} = 1 \mid Y = 1, \mathbf{X} \in \mathcal{T}_1(\varepsilon)\right). \tag{105}$$

In all experiments, we report results for $\varepsilon \in \{0.10, 0.05, 0.01\}$, corresponding to the $90\%$, $95\%$, and $99\%$ detection-class tail regions.

**Interpretation.** Under elliptically contoured $\alpha$-stable models, large values of $r_1(\mathbf{x})$ correspond to extreme realizations of the latent radial variable, where likelihood decay is governed by the stability index $\alpha$. TailRecall therefore directly evaluates detection performance in the regime where heavy-tailed modeling is statistically meaningful.

## I.2. Precision–Recall AUC (PR-AUC)

We additionally report the area under the precision–recall curve (PR-AUC), a standard metric for imbalanced binary classification.

Let $\tau \in \mathbb{R}$ denote a decision threshold applied to a real-valued classifier score $s(\mathbf{x})$. For each $\tau$, define precision and recall as

$$\text{Precision}(\tau) = \mathbb{P}(Y = 1 \mid s(\mathbf{X}) \geq \tau), \tag{106}$$

$$\text{Recall}(\tau) = \mathbb{P}(s(\mathbf{X}) \geq \tau \mid Y = 1). \tag{107}$$

The precision–recall curve is obtained by varying $\tau$ over its full range. The PR-AUC is defined as

$$\text{PR-AUC} = \int_0^1 \text{Precision}(\text{Recall}^{-1}(r)) \, dr, \tag{108}$$

which corresponds to the area under the precision–recall curve.

**Role and limitations.** PR-AUC is sensitive to class imbalance and provides a global summary of precision–recall trade-offs across thresholds. However, it aggregates performance over the entire feature space and is therefore dominated by behavior in high-density regions. As a result, improvements in tail-region detection may yield only modest gains in PR-AUC.

**Complementarity.** For this reason, PR-AUC and TailRecall serve complementary roles in our evaluation: PR-AUC assesses overall ranking quality, while TailRecall isolates performance on rare, large-deviation events characteristic of heavy-tailed data.

# J. Real-World Experiments: Full Results

This appendix provides complete results for all real-world datasets evaluated in Section 8.

## J.1. HTRU2: Pulsar Detection

**Dataset.** The HTRU2 dataset (Lyon, 2015) contains 17,898 candidates from the High Time Resolution Universe Survey, with 1,639 confirmed pulsars (9.2%) and 16,259 non-pulsars. Each candidate is described by 8 features: mean, standard deviation, excess kurtosis, and skewness of both the integrated pulse profile and the DM-SNR curve.

**Preprocessing.** StandardScaler normalization.

**Diagnostic output.**

- Estimated $\alpha$: Class 0 (non-pulsar) = 1.50, Class 1 (pulsar) = 2.00, Average = 1.73
- Scale ratio: 3.2, Determinant ratio: 244
- Heavy-tail signals: 3/3 (outlier rate 8.4%, mean-median shift 0.53)
- Tyler safe threshold: $\alpha < 1.7$
- **Recommendation: Standard estimators** (mean + Ledoit–Wolf)

The diagnostic correctly identifies that Tyler's M-estimator would lose discriminative scale information due to the large determinant ratio, while standard estimators remain viable because the moderate tail index ($\alpha \approx 1.73$) does not severely corrupt sample mean and covariance.

*Table 22.* HTRU2 complete results (5-fold CV, mean $\pm$ std).

| Method | Accuracy | PR-AUC | Rec@P95 | F1 |
|---|---|---|---|---|
| Gaussian QDA | $96.77 \pm 0.30$ | $0.893 \pm 0.012$ | $0.707 \pm 0.031$ | $0.830 \pm 0.014$ |
| Stable (mean+LW) | $\mathbf{97.83 \pm 0.22}$ | $\mathbf{0.911 \pm 0.014}$ | $\mathbf{0.796 \pm 0.030}$ | $\mathbf{0.873 \pm 0.013}$ |
| Stable (smed+Tyler) | $93.85 \pm 0.35$ | $0.836 \pm 0.018$ | $0.572 \pm 0.042$ | $0.718 \pm 0.015$ |
| Stable (smed+LW) | $97.81 \pm 0.20$ | $0.910 \pm 0.013$ | $0.794 \pm 0.028$ | $0.872 \pm 0.012$ |

**Improvement over Gaussian QDA (mean+LW).**

- $\Delta$ PR-AUC: +2.0% ($p < 0.001$)
- $\Delta$ Recall@Precision$\geq$95%: +12.6% ($p < 0.001$)
- Error reduction: 32.5%

**Analysis.** The stable likelihood substantially improves high-precision detection: at 95% precision, Stable-QDA recalls 79.6% of pulsars compared to 70.7% for Gaussian QDA. Tyler's M-estimator performs poorly ($-89.3\%$ error reduction), confirming the diagnostic prediction.

### J.2. Credit Card Fraud Detection

**Dataset.** The Credit Card Fraud dataset (Machine Learning Group, Université Libre de Bruxelles, 2023) contains 284,807 transactions from European cardholders in September 2013, with 492 frauds (0.17%). Features are 28 PCA-transformed components plus transaction amount and time.

**Preprocessing.** StandardScaler normalization.

**Diagnostic output.**

- Estimated $\alpha$: Average = 1.73
- Scale ratio: 1.02, Determinant ratio: $1.14 \times 10^7$
- Heavy-tail signals: 3/3
- Tyler safe threshold: $\alpha < 1.6$
- **Recommendation: Standard estimators** (mean + Ledoit–Wolf)

*Table 23.* Credit Card Fraud complete results (5-fold CV, mean $\pm$ std).

| Method | Accuracy | PR-AUC | Rec@P95 | F1 |
|---|---|---|---|---|
| Gaussian QDA | $97.52 \pm 0.11$ | $0.488 \pm 0.042$ | $0.000 \pm 0.000$ | $0.105 \pm 0.009$ |
| Stable (mean+LW) | $\mathbf{98.83 \pm 0.05}$ | $\mathbf{0.739 \pm 0.031}$ | $0.000 \pm 0.000$ | $\mathbf{0.427 \pm 0.026}$ |
| Stable (smed+Tyler) | $91.80 \pm 0.25$ | $0.437 \pm 0.038$ | $0.000 \pm 0.000$ | $0.032 \pm 0.004$ |
| Stable (smed+LW) | $98.81 \pm 0.06$ | $0.738 \pm 0.032$ | $0.000 \pm 0.000$ | $0.423 \pm 0.027$ |

**Improvement over Gaussian QDA (mean+LW).**

- $\Delta$ PR-AUC: +51.4% ($p < 0.001$)
- Error reduction: 52.9%

**Analysis.** The massive determinant ratio ($> 10^7$) indicates extreme heteroscedasticity between fraud and legitimate transactions. Standard estimators preserve this discriminative information, yielding a 51% relative improvement in PR-AUC. Tyler's trace normalization destroys this signal, resulting in catastrophic performance degradation ($-226\%$ error reduction).

### J.3. Ionosphere: Radar Signal Classification

**Dataset.** The Ionosphere dataset (Sigillito et al., 1989) contains 351 radar returns from a phased array in Goose Bay, Labrador. The task is to classify returns as "good" (225 samples, showing evidence of structure in the ionosphere) or "bad" (126 samples, passing through without reflection). Each return is described by 34 continuous features.

**Preprocessing.** StandardScaler normalization.

**Diagnostic output.**

- Estimated $\alpha$: Average = 1.91
- Scale ratio: 1.18, Determinant ratio: N/A (high-dimensional)
- Heavy-tail signals: 2/3 (outlier rate 10.2%, mean-median shift 0.24)
- Tyler safe threshold: $\alpha < 1.8$
- **Recommendation: Standard estimators** (mean + Ledoit–Wolf)

**Improvement over Gaussian QDA (mean+LW).**

- $\Delta$ PR-AUC: +2.6% ($p = 0.029$)
- $\Delta$ Recall@Precision$\geq$95%: +28.9%

*Table 24.* Ionosphere complete results (5-fold CV, mean $\pm$ std).

| Method | Accuracy | PR-AUC | Rec@P95 | F1 |
|---|---|---|---|---|
| Gaussian QDA | 87.46 $\pm$ 4.12 | 0.965 $\pm$ 0.017 | 0.724 $\pm$ 0.112 | 0.897 $\pm$ 0.032 |
| Stable (mean+LW) | **91.74 $\pm$ 3.28** | **0.990 $\pm$ 0.008** | **0.933 $\pm$ 0.067** | **0.934 $\pm$ 0.024** |
| Stable (smed+Tyler) | 92.03 $\pm$ 2.95 | 0.983 $\pm$ 0.012 | 0.902 $\pm$ 0.078 | 0.937 $\pm$ 0.021 |
| Stable (smed+LW) | 91.46 $\pm$ 3.41 | 0.989 $\pm$ 0.009 | 0.924 $\pm$ 0.071 | 0.932 $\pm$ 0.025 |

- Error reduction: 34.1%

**Analysis.** Despite moderate tails ($\alpha \approx 1.91$), the stable likelihood provides significant improvement. The high-dimensional setting ($d = 34 > n_{\min} = 126$) makes covariance estimation challenging; Ledoit–Wolf shrinkage combined with the stable likelihood's robustness to outliers yields consistent gains.

### J.4. Weekly Stock Returns

**Dataset.** The Weekly dataset (James et al., 2013) contains 1,089 weekly percentage returns for the S&P 500 from 1990 to 2010. The task is to predict market direction (up/down) from 8 features: lag returns (Lag1–Lag5), trading volume, and two technical indicators.

**Preprocessing.** StandardScaler normalization.

**Diagnostic output.**

- Estimated $\alpha$: Average = 1.67
- Scale ratio: 1.08, Determinant ratio: 1.16
- Heavy-tail signals: 2/3 (outlier rate 8.1%, $\alpha < 1.8$)
- Tyler safe threshold: $\alpha < 2.0$ (small det ratio)
- **Recommendation: Robust estimators** (spatial median + Tyler)

*Table 25.* Weekly complete results (5-fold CV, mean $\pm$ std).

| Method | Accuracy | PR-AUC | Rec@P95 | F1 |
|---|---|---|---|---|
| Gaussian QDA | 94.12 $\pm$ 1.23 | 0.993 $\pm$ 0.004 | 0.980 $\pm$ 0.015 | 0.956 $\pm$ 0.009 |
| Stable (mean+LW) | 95.04 $\pm$ 1.08 | 0.995 $\pm$ 0.003 | 0.979 $\pm$ 0.016 | 0.963 $\pm$ 0.008 |
| Stable (smed+Tyler) | **96.05 $\pm$ 0.95** | **0.996 $\pm$ 0.003** | 0.979 $\pm$ 0.014 | **0.971 $\pm$ 0.007** |
| Stable (smed+LW) | 95.87 $\pm$ 1.01 | 0.996 $\pm$ 0.003 | **0.981 $\pm$ 0.013** | 0.969 $\pm$ 0.008 |

**Improvement over Gaussian QDA (smed+Tyler).**

- $\Delta$ PR-AUC: +0.3% ($p = 0.008$)
- Error reduction: 32.8%

**Analysis.** This is the one dataset where robust estimators outperform standard ones, as predicted by the diagnostic. The small determinant ratio (1.16) means Tyler's trace normalization loses minimal discriminative information, while the heavier tails ($\alpha \approx 1.67$) benefit from robust location estimation.

### J.5. Additional Datasets

We also evaluated two datasets where improvements were modest and not statistically significant, included here for completeness.

#### J.5.1. PIMA INDIANS DIABETES

**Dataset(Smith et al., 1988).** 768 samples, 8 features, 35% positive class.

**Diagnostic.** $\alpha \approx 1.96$ (light tails), det ratio = 17, recommendation = standard.

**Analysis.** Light tails ($\alpha \approx 1.96$) limit the benefit of the stable likelihood. Small sample size ($n = 768$) yields high variance and non-significant improvements.

*Table 26.* Diabetes results (5-fold CV).

| Method | Accuracy | PR-AUC | $p$-value |
|---|---|---|---|
| Gaussian QDA | 73.4% | 0.647 | — |
| Stable (mean+LW) | 75.8% | 0.669 | 0.491 |
| Stable (smed+Tyler) | 74.5% | 0.660 | 0.749 |

### J.5.2. CUSTOMER CHURN

**Dataset**(Orange Telecom, 2010). 3,150 samples, 10 features, 27% positive class.

**Diagnostic.** $\alpha \approx 1.96$ (light tails), det ratio = 8,845, recommendation = standard.

*Table 27.* Churn results (5-fold CV).

| Method | Accuracy | PR-AUC | $p$-value |
|---|---|---|---|
| Gaussian QDA | 83.1% | 0.751 | — |
| Stable (mean+LW) | 83.7% | 0.698 | 0.009 |
| Stable (smed+Tyler) | 85.2% | 0.748 | 0.682 |

**Analysis.** Light tails and mixed results across metrics. The stable likelihood improves accuracy but not PR-AUC, suggesting the benefit is primarily in threshold-dependent classification rather than ranking.

### J.6. Summary

Table 28 summarizes results across all six datasets.

*Table 28.* Summary of real-world experiments. Stable-QDA achieves significant improvements on 4/6 datasets.

| Dataset | $n$ | $d$ | $\hat{\alpha}$ | Best Method | Err. Red. | |
|---|---|---|---|---|---|---|
| HTRU2 | 17,898 | 8 | 1.73 | mean+LW | 32.5% | ✓ |
| Credit Card | 284,807 | 28 | 1.73 | mean+LW | 52.9% | ✓ |
| Ionosphere | 351 | 34 | 1.91 | mean+LW | 34.1% | ✓ |
| Weekly | 1,089 | 8 | 1.67 | smed+Tyler | 32.8% | ✓ |
| Diabetes | 768 | 8 | 1.96 | mean+LW | 8.8% | ✗ |
| Churn | 3,150 | 10 | 1.96 | smed+Tyler | 12.2% | ✗ |

**Key observations:**

1. Stable-QDA significantly outperforms Gaussian QDA on 4/6 datasets ($p < 0.05$).
2. Average error reduction on significant datasets: 38.1%.
3. The diagnostic correctly predicts the optimal estimator on 4/6 datasets.
4. Datasets with light tails ($\alpha > 1.9$) show modest, non-significant improvements.
5. Large determinant ratios ($> 100$) reliably indicate that standard estimators should be preferred over Tyler.

