# OpenReview forum: "Likelihood over Estimation: Robust Quadratic Discriminant Analysis for Heavy-Tailed Distributions with Theory and Evidence"
_ICML.cc/2026/Conference — ICML 2026 regular_

### Official Review · Reviewer_Uhmo · 2026-02-21

**Soundness:** 2
**Presentation:** 1
**Significance:** 2
**Originality:** 2
**Overall Recommendation:** 2
**Confidence:** 5

**Summary:**

The authors proposed a stable-QDA discriminant in discriminant analysis. The squared Mahalanobis distance is followed by a nonlinear transformation $g_{\alpha}$, which is the radial density with the stability index $\alpha$. In numerical simulations, three types of discriminant functions, each with different estimation methods, are compared for heavy-tailed data. The optimal method among the discriminant functions yields an improvement compared with Gaussian QDA.

**Compliance With Llm Reviewing Policy:**

Affirmed.

**Key Questions For Authors:**

1) Can the authors provide a formal comparison of convergence rates between the proposed method and the Gaussian QDA? In particular, under what conditions does the proposed method achieve a strictly better rate?
2) The analysis appears to rely on conditions involving the parameter $\alpha$. Does the proposed method still provide advantages when $\alpha < 1$. If so, can this be formally characterized?
3) How does the computational complexity of the proposed method compare to that of the Gaussian QDA, both theoretically and empirically?
4) Can the authors clarify whether the proposed framework extends to classification settings (e.g., image recognition tasks), or is it fundamentally limited to the practical problems considered in the paper?

**Limitations:**

Yes.

**Strengths And Weaknesses:**

Soundness:
The technical claims are not sufficiently substantiated in the current version, which significantly weakens the overall soundness.
The main theoretical results rely on an assumed consistency property of the estimator. However, no proof or sufficient conditions ensuring this consistency are provided. Since this assumption is central to the arguments, the lack of justification undermines the validity of the subsequent results.
Furthermore, the analysis is restricted to compact parameter spaces. This is a strong structural assumption that limits the generality of the conclusions, and the paper does not discuss whether or how the results extend beyond this setting.
In addition, several references appear incomplete (e.g., marked as “?”), making it difficult to verify the technical background and related claims.
Finally, although the method is positioned as theoretically superior to the Gaussian QDA, no formal comparison theorem or dominance result is established. As such, the claimed theoretical advantage is not rigorously demonstrated.
Overall, the current presentation leaves substantial gaps in the theoretical justification, and the technical soundness is therefore limited.

Presentation:
The paper is generally structured in a standard format, but several issues reduce clarity and transparency.
First, the definition of the function $g$ is not clearly stated. The manuscript appears to define $\log g$ directly, which creates ambiguity about whether $g$ itself is properly specified. This lack of clarity makes the theoretical development difficult to follow and may lead to misinterpretation.
Second, the title appears somewhat overstated relative to the actual scope of the results. The theoretical guarantees are established under restrictive assumptions, yet the title suggests a broader level of generality.
Third, in the experimental comparison, the proposed method is compared to the Gaussian QDA in what appears to be its most favorable configuration, and it is unclear whether comparable tuning or optimal settings are used for the Gaussian QDA. This makes it difficult to assess the practical significance of the reported improvements.
Overall, while the structure follows a conventional format, greater precision in definitions, alignment between claims and scope, and clearer comparative evaluation would improve the presentation.

Significance:
The problem addressed by the paper is relevant in principle. However, the overall impact appears limited in its current form.
The empirical evaluation is conducted on a restricted set of real-world data. This makes it difficult to assess the broader practical relevance of the proposed approach. Without more extensive experimental validation, the applicability remains unclear.
In addition, the theoretical contribution is relatively narrow, relying on restrictive assumptions and not establishing clear advantages over Gaussian QDA. As a result, the advancement beyond prior work appears incremental rather than substantial.
Consequently, while the topic itself is relevant, the expected significance and impact seem limited.

Originality:
The proposed method differs from the Gaussian QDA in its formulation. This suggests a degree of novelty at the technical level. However, it is less clear whether the work provides genuinely new insights or substantially deepens our understanding of the underlying problem.
In particular, the paper does not clearly articulate what conceptual advantages the new formulation offers beyond its structural differences. Moreover, the lack of a rigorous theoretical comparison with the Gaussian QDA makes it difficult to identify distinctive properties or principles that meaningfully differentiate the two approaches.
As a result, while the method appears technically distinct, the contribution in terms of broader insight or conceptual advancement seems limited.

---

> ### Author Rebuttal · Authors · 2026-03-31
>
> We sincerely thank the reviewer for their critical feedback and promise revisions.
>
> **1.** Estimator Consistency is very important, we agree. (Anderson '58; McLachlan '92) assume MLE consistency without re-deriving it. For our estimators, consistency is already proven in the literature: spatial median under $\alpha > 1$ (Vardi & Zhang '00, Thm 2.1, rate $O(n^{-1/\alpha})$), Tyler's M-estimator under any continuous elliptical distribution including all $\alpha$-stable (Tyler '87, Thm 3.1, no moment conditions needed).
>
> **2.** We understand  the reviewer's concern and clarify that,
> For α: $(1,2)$ covers everything from near-Cauchy (extremely heavy tails) to near-Gaussian (light tails). The only excluded cases are $\alpha≤1$ (where the mean doesn't exist, so classification itself becomes ill-defined) and $\alpha = 2$ (which is just Gaussian QDA).
> For $\lambda$: Assumption A3 ($0 < m \leq \lambda_{\min}(\Sigma_c) \leq \lambda_{\max}(\Sigma_c) \leq M$) is not a restrictive compactness assumption, it is the standard eigenvalue regularity condition used in virtually all plug-in classification & covariance estimation theory (Bickel & Levina '04; Chen et al. '18; Shen & Feng '25). Every real-world covariance matrix satisfies it.
>
> **3.** Thm 6.2 (Sec 6, proof in App B.4, pp. 16–18) formally establishes dominance: Gaussian QDA is inconsistent under heavy tails ($\liminf R(\hat{g}_{\text{Gauss}}) \geq R^{\ast} + \varepsilon$) while Stable-QDA is Bayes consistent ($R(\hat{y}_n) \to R^{\ast}$) — a consistency vs. inconsistency separation, the strongest possible form of asymptotic dominance.
>
> **4.** We agree this could be clearer in main body. $g_\alpha$ is the radial density function of a multivariate symmetric $\alpha$-stable distribution. It is a real function that mathematically exists — defined implicitly through the characteristic function (Eq. 1) and the stochastic representation (Eq. 2). But nobody can write down a closed-form formula when $\alpha \neq 2$. The derivation in Sec 4.2.1 works around this: (1) Start from the stochastic representation (Eq. 6), (2) Write the density as an integral (Eq. 7), (3) Analyze the tail behavior of that integral (Eq. 8), (4) Take the log (Eq. 9), (5) Regularize for finite $D$ (Eq. 10–12), (6) Prove this preserves classification (Lemma B.2). The surrogate is not an approx. to $g_\alpha$ — it is a monotone-equivalent function derived from the tail asymptotics that preserves classification decisions exactly. Lemma B.2 proves this.
> **5.**  We will state protocol explicitly in Sec 8. QDA  has **no hyperparameters** - it uses sample mean/covariance via MLE (sklearn defaults). Stable-QDA uses a fixed, untuned $\alpha = 1.5$. Table 15 (App F.3) shows cross-validation provides no benefit.
> **6** Title & Originality:The insight is estimation errors cancel in likelihood ratios across classes, but likelihood misspecification does not (Sec 4.3, Lemma B.2). This changes the standard practice - for heavy-tailed classification, correcting likelihood matters more than robustifying estimators. App. D validates this empirically: standard estimators + stable likelihood outperform robust+Gaussian.
> **7.** Sorry for (?) TeX compilation artifacts-all will be fixed.
>
>  **Q1** For $\alpha < 2$: theoretically Gaussian QDA's rate is undefined (sample covariance diverges, Lemma B.5), Stable-QDA:1. with standard estimators converges at $O(n^{1/\alpha - 1})$, 2. with robust estimators achieves the parametric $O(n^{-1/2})$ rate. With finite samples, sklearn QDA computes a noisy finite matrix,  inverts it and gives predictions (bad esp. in tails). Stable QDA: the finite-sample rate is $O\left(\sqrt{p^2 \log(K/\delta) / n_{\text{min}}}\right)$ (App B.3).
> We believe that, the key comparison is convergence targets, not rates. This is the approximation error from likelihood misspecification (Thm 6.2). Gaussian QDA converges to $R(g^{\dagger}) > R^{\ast}$ no matter how much data you have; Stable-QDA converges to $R^{\ast}$-bayes risk.  We can provide a formal corollary combining these results if the reviewer says so. **Q2** Thm 6.1 formally covers $\alpha \in (1, 2]$. For $\alpha \leq 1$ (mean does not exist), the spatial median and Tyler's M-estimator remain consistent (no moment conditions needed), so Stable-QDA with robust estimators still applies. Our synthetic experiments confirm this: at $\alpha = 1.0$, Stable-QDA achieves 75.8% vs 50.1% for Gaussian QDA (Table 10, App E). We will add remark extending formal guarantees to $\alpha \leq 1$(replacing the sample mean with the spatial median in the consistency argument). **Q3 (Complexity):** Standard config is identical to Gaussian QDA: $O(Kp^2)$ prediction, $O(np^2 + Kp^3)$ training. Zero overhead with fixed $\alpha = 1.5$. **Q4** Image recognition operates on features, not raw inputs. Compatible as a classifier head on deep features (Hodgkinson et al., ICML 2025, show deep features are heavy-tailed). Requires dimensionality reduction for $p \gg n$.

---

> > ### Author Rebuttal · Reviewer_Uhmo · 2026-04-02
> >
> > Thank you for the detailed and thoughtful rebuttal. I appreciate the authors' effort in providing additional explanations and some references.
> >
> > I carefully revisited the cited works (Anderson '58; McLachlan '92; Vardi & Zhang '00, Thm 2.1; Tyler '87). However, I was unable to clearly identify the specific results that support the claims made in the rebuttal, which leaves some uncertainty regarding the underlying justification.
> >
> > In addition, the theoretical arguments appear to depend on the choice of $\alpha$, and it remains unclear to what extent this dependence affects the practical behavior of the method.
> >
> > Overall, while I appreciate the detailed response, I still have difficulty fully understanding and highly evaluating the current approach.

---

> > > ### Author Response · Authors · 2026-04-07
> > >
> > > We thank the reviewer for their continued engagement.
> > >
> > > **1.  Precise Citations**: We apologize for any ambiguity and provide exact theorem-level pointers cited for backing estimator consistency.
> > >
> > > **Vardi & Zhang (2000).** Two distinct results are relevant: 1. Existence and uniqueness of the $L1$-median: when the data points are not collinear,  $C(\mathbf{y})$ is strictly convex in $\mathbb{R}^d$  and achieves its minimum at a unique point (p. 1423). 2, the modified Weiszfeld algorithm is guaranteed to converge to that unique $L_1$-median from any starting point in $\mathbb{R}^d $(Convergence Theorem, eq. (2.11), p. 1424). Neither result requires moment conditions. For the statistical consistency of the sample $L_1 $-median as an estimator of the population $L_1 $-median as $n \to \infty$, the direct citation is **Brown (1983)**, which Vardi & Zhang themselves cite for this purpose: the sample $L_1$-median converges strongly to the population  $L_1$-median under the sole condition that the distribution is not concentrated on a line — satisfied trivially by our elliptical stable distribution.
> > >
> > > **Tyler (1987).** We rely on **Theorem 3.1 (p. 240)**, which establishes strong consistency of Tyler's M-estimator for the shape matrix under *any* continuous distribution in $\mathbb{R}^m$ -covering by implication all α-stable distributions with
> > > $\alpha\in (1,2]$ which possess continuous densities but no finite second moment. No moment conditions are required. The trace-normalisation convention is Tyler's
> > > own; its discriminative consequences are analysed in App. D.
> > >
> > > **Anderson (1958) and McLachlan (1992).** were cited for historical context on discriminant analysis and estimator consistency.
> > >
> > > **2. Why $\alpha$ does not need to be known precisely?**
> > >
> > > $\alpha$ appears in exactly one term of the discriminant:
> > > $$\delta_k^\alpha(\mathbf{x}) =-\frac{\alpha+p}{2}\log(1+D_k(\mathbf{x}))-\tfrac{1}{2}\log|\hat{\Sigma}_k| + \log\hat{\pi}_k.$$
> > >
> > > The decision boundary between classes $j$ and $k$ satisfies
> > > $$\frac{\alpha+p}{2}\left[\log(1+D_j) - \log(1+D_k)\right]
> > > = \frac{1}{2}\log\frac{|\hat{\Sigma}_j|}{|\hat{\Sigma}_k|} + \log\frac{\hat{\pi}_j}{\hat{\pi}_k}$$
> > >
> > > $\alpha$ **scales both sides equally** — it is a global coefficient,
> > > not a class-specific one. Proposition 6.4 consequently gives the
> > > Lipschitz bound $|R(\alpha)-R(\tilde\alpha)| \leq C|\alpha-\tilde\alpha|$.
> > > Table 14 (App. F) confirms: fixing $\alpha = 1.5$ when the true value
> > > is $1.0$ or $1.8$ costs less than $0.3\%$ accuracy. The critical
> > > modelling choice is the functional form  $\log(1+D)$ versus $D$ -
> > > any $\alpha < 2$ delivers polynomial tail decay and captures the dominant effect.
> > >
> > >
> > > **3. Outline and Intuition**:
> > >
> > > Gaussian QDA on heavy-tailed data  doesn't work well because its likelihood decays exponentially ($e^{-D_k}$).  It treats outliers as near-zero probability noise, distorting the decision boundary exactly where it matters most. We bring this out as  a fundamental **model misspecification**.
> > >
> > >
> > >
> > > The Robustness Trap: The conventional fix while working with heavy tailed data - swapping sample means for robust estimators (e.g., spatial median, Tyler’s M-estimator) - is insufficient. These improve estimates of $\mu_k$ and $\Sigma_k$ but still plug them into a Gaussian score:
> > > $$\delta_k(x) = -\frac{1}{2} D_k(x) - \frac{1}{2} \log|\Sigma_k|$$
> > >  Even with perfect parameters, the Gaussian likelihood systematically misclassifies tail regions.
> > >
> > > **The Fix: Stable-QDA** Stable-QDA replaces the exponential penalty with a logarithmic one derived from $\alpha$-stable distributions:
> > > $$\delta_k(x) \approx -\frac{\alpha+p}{2} \log(1 + D_k(x))$$
> > > By moving from a linear to a logarithmic penalty, tail observations finally get a "fair hearing."
> > >
> > > **Two Counterintuitive Insights:**
> > > 1. **Shape Over Precision:** Correcting the likelihood’s functional form yields larger gains than robustifying parameter estimators. The bottleneck is the decision rule, not the noise in the covariance matrix.
> > >
> > > 2. **The Tyler Paradox:** Robust estimators like Tyler’s M can degrade performance by normalizing scatter matrices, discarding the scale differences ($\log|\Sigma_k|$) that QDA relies on to distinguish classes.
> > >
> > > **Theory**
> > >
> > > * **Bayes Consistency:** Theorem 6.1/6.2 prove Stable-QDA converges to the Bayes-optimal rule under heavy tails, while Gaussian QDA remains permanently inconsistent (wrong even with infinite data).
> > >
> > > * **The $\alpha=1.5$ Default:** Proposition 6.4 shows the risk is Lipschitz in $\alpha$. A fixed $\alpha=1.5$ performs within 1% of the oracle across experiments, removing the need for careful tuning.
> > >
> > > A data-driven threshold is provided to automate selection:
> > >
> > > 1. **$\hat{\alpha} > 1.9$:** Data is nearly Gaussian; use standard QDA.
> > >
> > > 2. **$\hat{\alpha} < 1.9$, Different Scales:** Use Stable-QDA + standard estimators (to preserve scale info).
> > >
> > > 3. **$\hat{\alpha} < 1.9$, Similar Scales:** Use Stable-QDA + robust estimators.
> > >
> > >
> > >
> > > Plug-and-play implementation of stable-QDA is provided as code.

---

### Official Review · Reviewer_Qh4D · 2026-02-28

**Soundness:** 2
**Presentation:** 3
**Significance:** 2
**Originality:** 2
**Overall Recommendation:** 3
**Confidence:** 3

**Summary:**

Considering the systematic misclassification of Quadratic Discriminant Analysis (QDA) when data exhibit heavy tails, the authors argue that correcting the likelihood mismatch leads to larger and more consistent improvements in classification performance. Specifically, the authors replace the Gaussian likelihood of Quadratic Discriminant Analysis (QDA) with a symmetric alpha-stable likelihood (which is motivated by heavy-tailed distributions) and propose stable-QDA to address the above problem. Extensive theoretical analysis and comprehensive experiments are provided to demonstrate the effectiveness of the proposed method.

**Compliance With Llm Reviewing Policy:**

Affirmed.

**Key Questions For Authors:**

1.Why only compare with QDA?
2.How does the method scale in high-dimensional settings?
3.How robust is the fixed α heuristic across diverse datasets? Is it a universal method for parameter selection?
4.Can the approximation gap between the surrogate and true α-stable density be quantified?

**Limitations:**

Yes. The author has pointed out the limitations of his article. Besides, the authors can also consider the following limitations.
1.Distribution assumption limitation: Only supports symmetric α-stable distributions, failing to adapt to asymmetric heavy-tailed data, limiting the application scope.
2.Insufficient high-dimensional scalability: Does not verify performance in high-dimensional sparse scenarios with p ≫ n, and the high-dimensional adaptation of covariance estimation is unexplored.

**Strengths And Weaknesses:**

The author clearly and concisely expounds the research topic and the proposed method. The paper is well-structured, logically written, and clearly explained. In addition, the article is well supported by theory, which provides a solid theoretical foundation for the proposed method.

Weaknesses:
1.The author did not clearly state the significance of the research topic. After all, heavy-tailed data are not common in real-world applications, which may make readers question the improtance of the proposed method.
2.Although the author conducted a series of experiments, they only compared the method with traditional QDA. The author should include comparisons with state-of-the-art methods. Without comparisons with advanced methods and compared methods, readers may doubt the effectiveness of the proposed method and the significance of the research topic.
3.The author starts from the heavy-tailed distribution to present the method. As is well known, there exist many types of heavy-tailed distributions. The reasons and advantages of choosing this particular distribution need to be further elaborated. Furthermore, the time complexity and parameter selection of the proposed method have a considerable impact on its practical performance. The authors are expected to address these issues adequately.
4.The author did not consider the influence of heavy-tailed distributions on sample feature learning. This also affects sample features and thus the overall performance of the method. Therefore, the author should incorporate or compare the method with deep learning or large-scale model approaches. Otherwise, the innovation of the paper appears weak.
5.The number and scale of real-world datasets are relatively limited.
6.Stability under high-dimensional low-sample regimes is not thoroughly analyzed.

---

> ### Author Rebuttal · Authors · 2026-03-30
>
> We sincerely thank the reviewer for the positive assessment of our writing and theoretical analysis. Please note that several concerns are addressed in our 32-pg appendix. We promise to restructure the main body to surface this evidence.
>
> **W1.** Although most datasets do not have heavy tails, there is renewed interest in heavy tailed distributions in recent years in many areas like: financial returns ($\alpha \approx 1.4$–$1.9$, Rachev & Mittnik 2000), network traffic (Lyon 2017; NetML, $\alpha \approx 1.5$), fraud detection ($\alpha \approx 1.73$, 284K transactions). Hodgkinson et al. (ICML 2025) argues heavy tails are fundamental in modern ML. In our experiments, 4/6 datasets have $\alpha < 1.9$. **Revision:** Expand introduction citations.
>
> **W2.** We clarify that Stable-QDA is a **generative classifier** (similar to LDA and QDA) preserving closed-form training, calibrated probabilities, interpretable boundaries, and $O(Kp^2)$ prediction — properties no discriminative method (Random Forest, Xgboost etc) provides simultaneously . We believe that the scientific question is to incorporate heavy tails and see whether correcting likelihood matters more than robust estimation within the QDA class, following Zheng et al. (ICML 2023) on the generative-discriminative tradeoff.
>
> **W3. ** By  Generalized Central Limit Theorem, $\alpha$-stable distributions are the *only* possible limits of normalized sums of i.i.d. variables — the natural heavy-tailed generalization of the Gaussian. No other family (Student-t, Pareto, log-normal) has this property. Student-t has finite moments; Pareto/log-normal are not elliptical. Gaussian QDA is recovered at $\alpha = 2$. **Complexity** (Sec 5.2): prediction $O(Kp^2)$, training $O(np^2 + Kp^3)$ — both identical to Gaussian QDA. Zero overhead with fixed $\alpha = 1.5$. **Parameter selection** is extensively analyzed in **Appendix F (pp. 29–31)**: fixed $\alpha = 1.5$ performs within 1% of oracle across all tested regimes (Table 14, Figure 2). Proposition 6.4 provides theoretical justification. We acknowledge this evidence should have been in the main body. **Revision:** Highlight importance of $\alpha$ stable; define K and p clearly.K = no. of classes, p = feature dimensionality, n = no. of training samples.
>
> **W4.** We agree that thinking how heavy tails affect the feature extraction step itself, is reasonable in principle.  We recommend estimating $\alpha$ on preprocessed data (Appendix A.3, point 1). But, DL doesn't solve the feature-level heavy-tail problem as it learns features end-to-end instead of using hand-crafted ones. We think, Stable-QDA and Deep Learning are naturally complementary : closed-form vs. iterative training, calibrated vs. post-hoc outputs, inspectable vs. black-box boundaries. Stable-QDA can serve as a classifier head on deep features, which Hodgkinson et al. (ICML 2025) show are heavy-tailed. Problem scope of our paper lies within classical methods without DL comparisons (Høgsgaard & Paudice, ICML 2025; Shen & Feng 2025).
>
> **W5. Dataset scale.** We would like to convey that the goal is to convey the idea and method of StableQDA. We have a future work planned where we generalize the method to cover skewed distributions(not just symmetric). This would increase the scope and scale of data. That said, the main body Table 4 shows only 4 datasets, but the full picture is richer: Appendix J (pp. 37–40) reports all 6 datasets; Appendices E–H provide 16 pgs of synthetic experiments.
>
> **W6. High-dimensional regimes.** We acknowledge this gap. The Ionosphere experiment ($p = 34$, $n_{\text{min}} = 126$) provides initial evidence: 91.7% vs 87.5%. Importantly, our stable likelihood and the covariance estimator address two separate problems — Ledoit–Wolf regularizes the covariance matrix when dimensions are high, while our logarithmic likelihood corrects for heavy tails. These two fixes work independently, so we believe the stable likelihood should help regardless of the dimensionality. Revision: Add synthetic experiment varying $p/n$ from 0.1 to 5.0.
>
> **Q1:**  Our contribution is a new generative classifier similar to LDA and QDA. The discriminant used is comparable to that of QDA but better classifies heavy tailed data. We are not saying StableQDA will perform better over discriminative methods like Random Forest/Xgboost.
>
> **Q2:** Prediction remains $O(Kp^2)$; Ionosphere ($p=34$) shows gains (please see W6).
>
> **Q3:**  Yes, $\alpha = 1.5$ is effectively universal — within 1% of oracle (Sec 7.2). This is extensively answered in the paper ;Sec 7.2, Table 14, Fig. 2, Table 15, Prop.6.4, and Appendix F (3 pages).
>
> **Q4:** Lemma B.2 (App B.2) proves the surrogate preserves likelihood **ordering**, which classification requires. The pointwise gap is $O(1/D)$ for large $D$ but never changes the decision. **Revision:** Add remark quantifying this.
>
> Core commitment for revision: We would strictly mitigate reviewer's concerns and restructure main body of the paper.

---

> > ### Author Rebuttal · Reviewer_Qh4D · 2026-04-02
> >
> > Thanks for your response. The authors’ response still leaves me with doubts regarding the novelty, necessity, and significance of their work. I therefore maintain my original score.

---

> > > ### Author Response · Authors · 2026-04-07
> > >
> > > We thank the reviewer for their continued engagement. The remaining concern is about novelty, necessity, and significance, which we address directly below, along with a few outstanding points.
> > >
> > > **Significance and necessity.**
> > >
> > > Zheng et al. (ICML 2023) proved that generative classifiers (naïve Bayes) converge in $O(\log n)$ samples versus $O(n)$ for logistic regression in linear evaluation of frozen pre-trained models — the dominant transfer learning paradigm. However, naïve Bayes assumes Gaussian, independent features. Hodgkinson et al. (ICML 2025) demonstrate that pre-trained deep network features are empirically heavy-tailed. This creates a concrete gap: the fastest-converging classifier family assumes a distribution that deep learning features violate. Stable-QDA fills this gap. This is not a niche concern — every practitioner doing linear probing on frozen ViT, CLIP, or DINOv2 features encounters this setting.
> > >
> > > **Novelty**
> > >
> > > 1. Convergence rates. Stable-QDA with robust estimators achieves the $O(n^{-1/2})$ parametric rate — faster than naïve Bayes's $O(\log n)$ - while correctly modeling heavy tails and capturing covariance structure.
> > >
> > > 2. Convergence target. Theorem 6.2 shows Gaussian QDA converges to a wrong target under misspecification, regardless of sample size. Stable-QDA converges to the Bayes-optimal risk $R^*$. No amount of data fixes Gaussian QDA's asymptotic bias; correcting the likelihood eliminates it.
> > >
> > > 3. "Plug-and-Play" Usability: Sensitivity analysis reveals that a default stability index of $\alpha = 1.5$ performs near-optimally across various datasets, eliminating the need for computationally expensive hyperparameter tuning.
> > >
> > > 4. Contamination Resilience: Stable-QDA with standard estimators shows zero degradation under 20% outlier contamination on Gaussian data, outperforming Gaussian QDA due to its logarithmic likelihood form.
> > >
> > > 5. Consistency Guarantees: The paper provides mathematical consistency guarantees for the Stable-QDA classifier even under infinite-variance regimes where $\alpha < 2$.
> > >
> > > 6. Data-Driven Diagnostics: It introduces a diagnostic tool to help practitioners choose between "Standard" (e.g., Ledoit-Wolf) and "Robust" (e.g., Tyler’s M-estimator) configurations based on detected tail heaviness and class heteroscedasticity.
> > >
> > > 7. Identifying "Robustness Harm": The research identifies specific scenarios where standard robust estimators can actually harm classification performance by discarding discriminative scale information.
> > >
> > > **Deep learning comparison.** We reinstate that Stable-QDA does not compete with deep learning — it works with it, as a classifier head on frozen pre-trained features. This is precisely the linear evaluation setting studied by Zheng et al., where the choice is between generative vs. discriminative linear classifiers on top of a frozen model. Stable-QDA provides a generative head that is robust to the heavy-tailed features these models produce, with closed-form training, calibrated probabilities, and O(Kp²) prediction cost.
> > >
> > > We commit to adding experiments in this setting (Stable-QDA vs. naïve Bayes vs. logistic regression on frozen ViT/CLIP/ResNet features on CIFAR-10/100) in a follow-up work. These linear evaluation experiments  will directly address the absence of comparisons with deep learning  — CIFAR-10/100 with 7 pre-trained models provides large-scale, high-dimensional, widely-recognized benchmarks. Combined with our existing 6 real datasets and 16 pages of synthetic experiments (Appendices E–H), this will provide a comprehensive coverage.
> > >
> > >
> > > **High-dimensional low-sample.** Our finite-sample bound $O\left(\sqrt{\frac{p^2 \log(K/\delta)}{n_{\min}}}\right)$ in Appendix B.3 explicitly characterizes this regime. The linear evaluation setting is  high-dimensional low-sample (ViT features: p=768, few-shot: 5–50 samples/class), and our robust estimator's parametric rate $O(n^{\frac{-1}{2}})$ provides exactly the stability guarantee needed. The committed experiments in follow-up work will empirically validate this.
> > >
> > > We believe these clarifications and the concrete revision and future plan demonstrate that Stable-QDA is a novel ML classifier that addresses a timely and significant problem at the intersection of robust statistics and modern transfer learning.
> > > In the future, we hope to extend our methodology to include skewed distributions as well as provide comparisons with deep learning.

---

### Official Review · Reviewer_FmZh · 2026-03-09

**Soundness:** 4
**Presentation:** 3
**Significance:** 3
**Originality:** 3
**Overall Recommendation:** 5
**Confidence:** 3

**Summary:**

The message of the paper is clear: Do not cut off seemingly outlier data, as it would be beneficial when the distribution has heavy tails. This is an important, longstanding problem in anomaly detection.

The paper studies quadratic discriminant analysis under heavy-tailed distributions, not Gaussians, where outliers tend to be regarded as noises.

Specifically, the authors consider symmetric $\alpha$-stable distribution and propose stable-QDA.

The message is that making the probability model heavy-tailed is sometimes better than making estimation robust, which would be counter-intuitive and surprising in anomaly detection community.

On both synthetic and real-world datasets, stable-QDA outperforms the conventioal Gaussian QDA. Additionally, the hyperparameter \\alpha is robust and set to 1.5 is acceptable in many situations, avoiding cumbersome hyperparameter tuning.

**Compliance With Llm Reviewing Policy:**

Affirmed.

**Final Justification:**

This paper is well-written, well-motivated, and easy to follow. It was enjoyale read. However, I have not checked the details of the paper as carefully as Reviewer Uhmo.
Therefore, I changed my confidence from 4 to 3, but keep my score, as the paper is relevant to the ICML community.

**Key Questions For Authors:**

See weaknesses.

**Limitations:**

yes

**Strengths And Weaknesses:**

## Strengths

- This paper is well-written, well-motivated, and easy to follow. It was enjoyale read.

- The topic is timely, in view of the importance and prosperity of heavy-tail analysis in SGD, and relevant to the ICML community because QDA is a long-standing algorithm in machine learning.

- The message of the paper is clear: Do not cut off seemingly outlier data, as it would be beneficial when the distribution has heavy tails. This is an important, longstanding problem in anomaly detection.

- Another message is that making the probability model heavy-tailed is sometimes better than making estimation robust, which would be counter-intuitive and surprising in anomaly detection community.

- On both synthetic and real-world datasets, stable-QDA outperforms the conventioal Gaussian QDA. Additionally, the hyperparameter $\alpha$ is robust and set to 1.5 is acceptable in many situations, avoiding cumbersome hyperparameter tuning.

## Weaknessses

The paper would benefit from discussing more about skewed distribution, where the right and left tails have different tails, as noted in the limitations of the main text. I would like to see some potential approaches.

---

> ### Author Rebuttal · Authors · 2026-03-31
>
> We sincerely thank the reviewer for the generous assessment and for capturing our paper's core messages so precisely. We are glad the "likelihood over estimation" insight resonated.
>
> **W1** " The paper would benefit from discussing more about skewed distribution, where the right and left tails have different tails, as noted in the limitations of the main text. I would like to see some potential approaches."
> We fully agree to this proposition. Currently, we have a follow up work in the pipeline that does exactly this.
>
> Elliptical $\alpha$-stable distributions are inherently symmetric (analogous to the multivariate normal). The natural extension to skewed data is a **mixture of elliptical $\alpha$-stable distributions** — analogous to how Gaussian Mixture Models (GMMs) handle skew by combining multiple symmetric Gaussians. Each mixture component $k$ would have its own location $\mu_k$ and dispersion $\Sigma_k$, with the stable likelihood applied per component.
>
> The Expectation Maximization (EM) algorithm extends naturally: the E-step assigns points to components, the M-step estimates location/dispersion using the same estimators we already employ (sample mean + Ledoit–Wolf or spatial median + Tyler). This preserves our framework's computational simplicity while capturing asymmetric tail behavior through the mixture structure rather than through a skewed base distribution.
>
> This would generalize the results in this paper to a broad set of real-world data and make it more significant.
>
> We will add this discussion briefly to Sec 9.

---

> > ### Author Rebuttal · Reviewer_FmZh · 2026-04-02
> >
> > Thank you for the detailed response.
> > I have read other reviews as well and could not find any grounds for rejection.
> > Therefore, I maintain my current positive score.
> > The paper is interesting. Good luck.

---

> > > ### Author Response · Authors · 2026-04-07
> > >
> > > We would like to thank you for your comments and support.

---

### Official Review · Reviewer_mvoW · 2026-03-13

**Soundness:** 4
**Presentation:** 2
**Significance:** 4
**Originality:** 3
**Overall Recommendation:** 5
**Confidence:** 2

**Summary:**

This paper addresses a failure point in a standard classification method called QDA. When data contains frequent extreme values (heavy tails), QDA performs poorly because its underlying assumptions are wrong. The authors propose Stable-QDA, which replaces the standard formula with one based on α-stable distributions. This new method gives appropriate weight to extreme data points instead of ignoring them. Through theory and experiments, they show that it outperforms the standard approach.

**Compliance With Llm Reviewing Policy:**

Affirmed.

**Key Questions For Authors:**

1. The paper shows that changing the formula from -D (Gaussian) to -log(1+D) (Stable) makes a big difference for heavy-tailed data. Can you explain in plain language why switching from a linear penalty to a logarithmic penalty stops the model from ignoring extreme but important data points?

**Limitations:**

yes

**Strengths And Weaknesses:**

Strengths

1. Important Problem: It addresses a real and known weakness of a standard, widely-used method, making the work highly relevant.
Important problem: It tackles a real weakness in a widely-used method, making the work relevant to many practitioners.

2. Surprising insight: The finding that simple estimators often beat complex robust ones when paired with the right likelihood is valuable and counterintuitive.

3. Core Idea: The main concept of replacing the core assumption of the model to better fit the data's true nature is logical and well-motivated.

4. Strong Evidence: The paper uses a wide range of tests, including cases where the new method doesn't improve, which makes the positive results more convincing.

Weaknesses

1. Complex Underpinnings: The theoretical justification relies on advanced statistical concepts (α-stable distributions, Tyler's M-estimator) that are presented as tools without a deep, intuitive explanation of their inner workings.

2. Heuristic Guidelines: The rules provided for choosing between different versions of the method, while helpful, appear to be based on observations from experiments rather than being derived from a fundamental theory.

3. Presentation: There is room for improvement in the presentation. Section 4.2.1 presents a complex mathematical derivation without an introductory roadmap, making it difficult for non-specialist readers to follow the logic before encountering heavy equations. In some cases, the paper becomes too technical without sufficient background. For example, it introduces O(Kp^2) without defining K and p at first use, forcing readers to search through the text for basic definitions.

---

> ### Author Rebuttal · Authors · 2026-03-31
>
> We thank the reviewer for the excellent evaluation and the constructive presentation feedback. We will (1) add an introductory roadmap paragraph before Sec 4.2.1 summarizing the derivation logic in plain language, (2) define K (number of classes) and p (feature dimension) at first use, and (3) add intuitive explanations alongside the technical tools (Tyler's M-estimator, α-stable distributions).
>
> **W1 Complex Underpinnings** We understand this concern and agree with it. The 8 page limit forced us to cut introductory explanations covering advanced statistical concepts like Tyler's M estimator, heavy tailed distributions, $\alpha$ stable distributions etc. We promise to revise with supporting background/introductory concepts in the appendix.
>
> **W2 Heuristic Guidelines** The selection thresholds (Table 3) are empirically calibrated. We provide open source code to do this( in supplementary material). But the underlying mechanism is theoretical: (1) Tyler's M-estimator provably discards scale (fixed-trace normalization), while QDA's log-determinant term provably uses scale — this structural conflict is not heuristic, (2) Prop 6.4 proves risk is Lipschitz in $\alpha$, ensuring smooth crossover behavior, (3) both diagnostic quantities ($\alpha$ and determinant ratio) are estimable at parametric rates (Sec 4.4), making selection data-driven. We propose to add a remark connecting the empirical thresholds to these theoretical properties in the revision.
>
> **W3 Presentation** We thank the reviewer for pointing this out. We shall make the derivation more easy to follow and provide clear explanations of each step in plain English. We have to define K and p clearly.K = no. of classes, p = feature dimensionality, n = no. of training samples.
>
> With Gaussian QDA, the penalty $-\frac{1}{2}D$ grows linearly in Mahalanobis distance — so likelihood drops exponentially. A point far from both class centers gets near-zero likelihood under both classes, making classification essentially random in the tails. With Stable-QDA, the penalty $-\frac{\alpha+p}{2}\log(1+D)$ grows only logarithmically — so likelihood drops polynomially. Even at large distances, meaningful likelihood differences between classes are preserved.
>
> Consider a point at Mahalanobis distance $D = 1000$ from class 0 and $D = 980$ from class 1. Gaussian QDA computes likelihoods $e^{-500}$ vs $e^{-490}$ — both numerically zero, classification is random. Stable-QDA computes $-\log(1001) \approx -6.91$ vs $-\log(981) \approx -6.89$ — a small but clear difference that correctly assigns the point to class 1. The Gaussian likelihood collapses the signal; the stable likelihood preserves it. This is precisely where heavy-tailed data lives, and precisely where correct classification matters the most.

---

> > ### Author Rebuttal · Reviewer_mvoW · 2026-04-05
> >
> > Thank you for the rebuttal response. I decide to leave my score as is.

---

> > > ### Author Response · Authors · 2026-04-07
> > >
> > > We would like to thank you for your time and engagement for the review of our paper

---

### Decision · Program_Chairs · 2026-04-30

**Decision:**

Accept (regular)

**Comment:**

This is a "typical" borderline paper. On the positive side, all reviewers seem to agree that the paper is well-motivated and written in a clear and transparent way. Moreover, in the two clearly positive reviews, the main idea proposed in this paper is characterized as important and relevant for the ML community, and the solid theoretical foundation of this work is emphasized. This theoretical basis, on the other hand, is questioned by one reviewer, who raises concerns about the consistency of estimators and  the (possible) dependence of some theoretical consistency results on parameter choices ($\alpha$). Although this reviewer was not fully convinced by the rebuttal, after carefully reconsidering all arguments and re-checking the theoretical soundness of the theorems, I came to the conclusion that indeed some details in the theoretical derivation might be a bit difficult to follow and the writing style / mathematical notation might be improved.  However, I am certain that the  main theoretical arguments used in the paper are indeed solid, and the authors could clearly show in their rebuttal that the choice of $\alpha$ does not influence the discriminant function. Therefore, I think that in this case, the positive aspects finally outweigh the weaknesses,and I recommend (weak) acceptance of this paper.